# OSAQ: Outlier Self-Absorption for Accurate Low-bit LLM Quantization

Zhikai Li [1]  Zhen Dong [2]  Xuewen Liu [1]  Jing Zhang [1]  Qingyi Gu [1]

## Abstract

Large Language Models (LLMs) have demonstrated remarkable capabilities. However, their massive parameter scale leads to significant resource consumption and latency during inference. Post-training weight-only quantization offers a promising solution by reducing model size and accelerating token generation through alleviating the memory-bound issue. Nevertheless, the presence of inherent systematic outliers in weights continues to be a major obstacle. While existing methods, such as scaling and rotation, attempt to address this issue, the performance remains unsatisfactory. In this paper, we propose Outlier Self-Absorption Quantization (OSAQ), which performs additive weight suppression guided by the second-order low-rank property for low-bit weight-only quantization of LLMs. Specifically, we observe that the Hessian exhibits low-rank consistency across different inputs, with certain directions consistently showing vanishing curvature. Leveraging this property, we identify a stable null space of the Hessian and then construct an additive weight transformation by linearly combining the vectors within this null space, thereby suppressing weight outliers without affecting the task loss. This additive transformation can be absorbed into the weights offline, requiring no inter-layer transformations and introducing no inference overhead. Moreover, the construction is efficiently achieved by a closed-form solution, without resource-intensive training or iterative procedures. Extensive experiments demonstrate that OSAQ effectively suppresses outliers and enhances low-bit quantization performance. For instance, in 2-bit quantization, OSAQ, when integrated with GPTQ, achieves over 40% lower perplexity compared to vanilla GPTQ.

[1]Institute of Automation, Chinese Academy of Sciences [2]University of California, Santa Barbara. Correspondence to: Qingyi Gu <qingyi.gu@ia.ac.cn>.

*Proceedings of the 43rd International Conference on Machine Learning*, Seoul, South Korea. PMLR 306, 2026. Copyright 2026 by the author(s).

## 1. Introduction

Large Language Models (LLMs) exhibit exceptional understanding and generation capabilities (Zhao et al., 2023; Zhang et al., 2022; Touvron et al., 2023), achieving state-of-the-art performance across a wide range of complex tasks such as reasoning, knowledge integration, and multi-modal interaction. Despite these advances, their success comes at the expense of enormous computational and memory demands (Xiao et al., 2023; Frantar & Alistarh, 2023), which translate into high deployment costs, significant inference latency, and large energy consumption. These inefficiencies pose serious challenges for real-world applications in resource-constrained or latency-sensitive environments, restricting the accessibility of LLMs (Zhou et al., 2024). As a result, reducing model complexity has become a topic of extensive research and growing interest.

Post-training quantization (PTQ), which discretizes model parameters into low-precision values without re-training or fine-tuning, is a promising direction for model compression (Nagel et al., 2020; Li et al., 2021). In the context of LLMs, the decoding process is often constrained by the memory wall (Gholami et al., 2024; Yuan et al., 2024), which severely limits memory access efficiency. Weight-only quantization has thus been widely investigated as an effective solution to this bottleneck (Wan et al., 2023; Lang et al., 2024). Nevertheless, model weights typically contain systematic outliers with large magnitudes, which significantly hinder quantization performance, particularly in low-bit settings. To this end, a variety of approaches have been proposed. For instance, GPTQ (Frantar et al., 2022) compensates for quantization errors of outliers by leveraging approximate second-order Hessian information. AWQ (Lin et al., 2023) mitigates the effects of outliers by scaling the weights according to activation distribution features. QuIP (Chee et al., 2023) introduces an orthogonal rotation of weight matrices to further suppress the presence of outliers. On this basis, several works have further invested in handling outliers within the scaling (Shao et al., 2023) and rotation (Ashkboos et al., 2024; Liu et al., 2024) paradigm.

Despite these efforts, the performance of low-bit quantization is still far from satisfactory, suggesting that relying solely on a multiplicative transformation paradigm is fundamentally inadequate. Therefore, we are motivated to explore

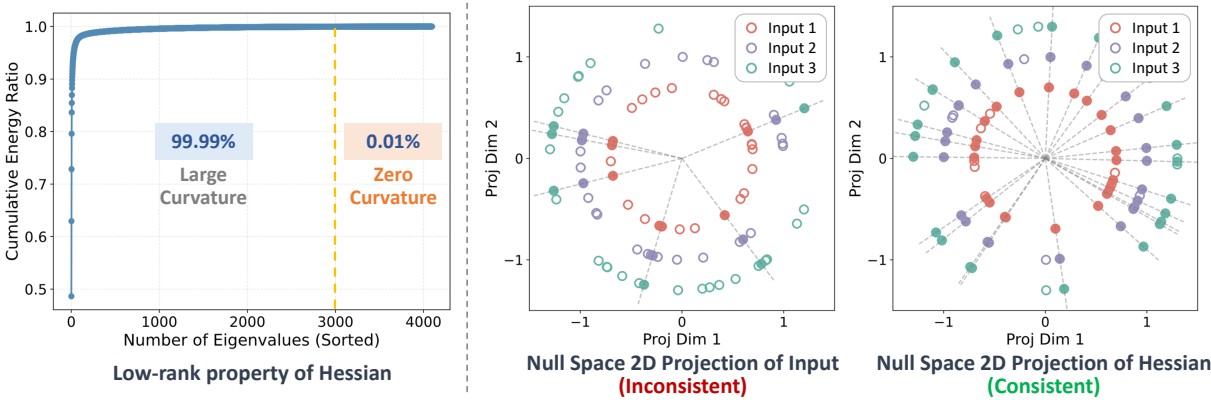

*Figure 1.* Visualization of the low-rank consistency of Hessian, using data from layer0.attn.k_proj module in LLaMA2-7B model. The *left* panel shows that a number of tail eigenvalues together contribute only about 0.01% of the total energy, indicating a pronounced low-rank property. The *right* panel compares the null space of the input and Hessian, where the high-dimensional vectors are projected into a 2D space for visualization. It can be observed that although the directions within the input null space vary significantly across different samples, the directions within the Hessian null space remain highly consistent.

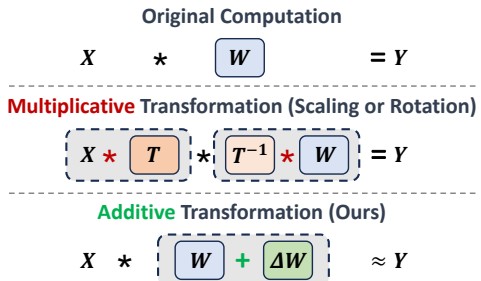

*Figure 2.* Illustration of additive transformation, which requires no adjustment to other layers while remaining approximately equivalent, thereby presenting a novel route for outlier suppression.

*whether there exist additional strategies for suppressing outliers beyond multiplicative transformations between adjacent layers.* Fortunately, we observe that the Hessian of the task loss with respect to a given layer's weights exhibits low-rank consistency across different inputs, as shown in Figure 1; in other words, the Hessian consistently shows negligible curvature in certain directions, referred to as the null space (Strang, 2022). This observation suggests the possibility of applying an additive transformation to the weights while keeping the second-order perturbation of task loss to zero, as shown in Figure 2. And more importantly, if such an additive adjustment can counteract the effect of outliers, it provides an effective strategy for suppressing them.

With the above insights, we propose Outlier Self-Absorption Quantization (OSAQ), an accurate low-bit weight-only quantization method for LLMs. OSAQ introduces an *additive weight transformation* beyond previous multiplicative approaches, offering a complementary and scalable scheme. Specifically, for a given layer, we first estimate its approximate Hessian and extract its null space by eigenvalue decomposition. Subsequently, the vectors in the null

space are linearly combined with coefficients to construct an additive transformation of the weights. Finally, we minimize the numerical range of the transformed weights by solving for the optimal combination coefficients, thereby effectively suppressing outliers. Note that enabled by the Softmax-∞ objective approximation, the solution of coefficients is obtained *directly in closed form*, without requiring resource-intensive training or iterative procedures. Our main contributions are summarized as follows:

- We propose an outlier self-absorption method that performs additive transformations by exploiting the low-rank properties of the second-order Hessian. It does not rely on inter-layer transformations and is complementary to scaling or rotation methods, further improving the performance of low-bit quantization for LLMs.

- The additive transformation is constructed by linearly combining the vectors in the null space of the Hessian. Thanks to the Softmax-∞ objective approximation, the combination coefficients can be derived in closed form to minimize the numerical range of weights.

- Extensive experiments are conducted on various models on a variety of tasks, and OSAQ significantly outperforms the existing methods in low-bit quantization. For instance, in 2-bit quantization, OSAQ achieves over 40% lower perplexity compared to vanilla GPTQ.

## 2. Related Work

**Post-Training Quantization.** Model quantization compresses neural networks by representing floating-point parameters with low-bit values (Gholami et al., 2022; Krishnamoorthi, 2018; Li & Gu, 2023). While many approaches adopt quantization-aware training (QAT), which requires

retraining on the full dataset to achieve strong performance (Choi et al., 2018; Esser et al., 2019), the QAT process is computationally expensive and time-consuming. In contrast, PTQ needs only a small number of samples for calibration, offering a more practical solution (Li et al., 2022; 2023). Previous PTQ methods, such as DFQ (Nagel et al., 2019), AdaRound (Nagel et al., 2020), and BRECQ (Li et al., 2021), perform well on small models, but their effectiveness diminishes on LLMs, where parameters exhibit pronounced outliers that amplify quantization difficulty and accumulate errors as model size grows. This has motivated increasing efforts to develop PTQ schemes for LLMs.

**Weight-Only Quantization for LLMs.** The PTQ methods for LLMs are categorized into two types: weight-activation quantization and weight-only quantization. The former needs to handle outliers in both weights and activations, with notable strategies including scaling (Xiao et al., 2023; Wei et al., 2022), rotation (Ashkboos et al., 2024; Liu et al., 2024), and permutation (Yuan et al., 2023; Lin et al., 2024), which rely on equivalent transformations between adjacent layers. However, the limited precision of activations imposes severe performance bottlenecks. To this end, due to the fact that the decoding efficiency is bounded by the memory access, weight-only quantization has received increasing interest. For outliers in weights, early approaches focused on explicit isolation or iterative compensation. For instance, LLM.int8() (Dettmers et al., 2022) and SqueezeLLM (Kim et al., 2023) isolate outliers independently. Afterwards, GPTQ (Frantar et al., 2022) performs error compensation iteratively using approximate Hessian, and MagR (Zhang et al., 2024) iteratively optimizes the infinite norm of the weights. In addition, methods based on equivalent transformations, such as scaling and rotation, have gained wide adoption. AWQ (Lin et al., 2023) mitigates the impact of outliers by scaling weights according to activation distributions, and QuIP (Chee et al., 2023) rotates the weights using orthogonal matrices to smooth and suppress outliers.

In this work, we focus on weight-only quantization and aim to explore a novel additive transformation based on Hessian's low-rank consistency as a new strategy for suppressing weight outliers. This approach complements existing inter-layer multiplicative transformations, providing a synergistic scheme that further enhances the performance of low-bit LLM quantization.

## 3. Preliminaries

**Model Quantization.** Quantization discretizes model parameters and represents them using low-precision numerical values (Gholami et al., 2022). To achieve this, the uniform quantizer is the most fundamental and hardware-friendly option, which is defined as follows:

$$\text{Quant: } \boldsymbol{w}^{(\mathbb{Z})} = \text{clip}\left(\left\lfloor\frac{\boldsymbol{w}}{s}\right\rceil + z, 0, 2^b - 1\right), \quad (1)$$

$$\text{De-Quant: } \hat{\boldsymbol{w}} = s\left(\boldsymbol{w}^{(\mathbb{Z})} - z\right) \approx \boldsymbol{w} \quad (2)$$

where $\boldsymbol{w}$ and $\boldsymbol{w}^{(\mathbb{Z})}$ denote the floating-point and quantized values, respectively, while the de-quantized value $\hat{\boldsymbol{w}}$ can approximately recover $\boldsymbol{w}$. Here, $\lfloor\cdot\rceil$ represents the rounding operation, and $b$ indicates the quantization bit precision. In this procedure, quantization scale $s \in \mathbb{R}^+$ and zero-point $z \in \mathbb{Z}$ are the key quantization parameters, which are determined by the upper bound $w_{\max}$ and lower bound $w_{\min}$ as follows:

$$s = \frac{w_{\max} - w_{\min}}{2^b - 1}, \quad z = \left\lfloor -\frac{w_{\min}}{s}\right\rceil \quad (3)$$

It can be observed that the scale $s$ is determined by the numerical range of $\boldsymbol{w}$, which directly affects the quantization resolution. However, in LLMs, the weights exhibit significant outliers, enlarging the range of $\boldsymbol{w}$ and consequently leading to reduced quantization resolution and accuracy.

**Multiplicative Transformation for Outlier Suppression.** To mitigate the outlier issue, various approaches have been proposed. A notable idea is based on scaling (Lin et al., 2023) or rotation (Chee et al., 2023), which is achieved through equivalent multiplicative transformations between adjacent layers, as follows:

$$\begin{aligned}(\boldsymbol{XW}_1)\boldsymbol{W}_2 &= (\boldsymbol{XW}_1)\boldsymbol{T}^{-1}\boldsymbol{TW}_2 \\ &= \boldsymbol{X}(\boldsymbol{W}_1\boldsymbol{T}^{-1})(\boldsymbol{TW}_2) = (\boldsymbol{XW}_1')\boldsymbol{W}_2' \end{aligned} \quad (4)$$

where $\boldsymbol{X}$ denotes the input, and $\boldsymbol{W}_1$ and $\boldsymbol{W}_2$ are the weights of two adjacent layers. $\boldsymbol{T}$ represents the transformation, which corresponds to scaling when it is a one-dimensional vector, and corresponds to rotation when it is a two-dimensional orthogonal matrix.

Despite certain improvements, the performance of low-bit quantization remains far from satisfactory, indicating that a single multiplicative paradigm is insufficient for handling outliers. Thus, we seek to explore a novel additive paradigm as a complementary approach to outlier suppression.

## 4. Observations and Insights

**Observations: Low-Rank Consistency of Hessian.** When the weights undergo a small additive perturbation, the second-order Taylor expansion of the task loss $\mathcal{L}$ for the weights is as follows:

$$\begin{aligned}&\mathbb{E}[\mathcal{L}(\boldsymbol{w} + \Delta\boldsymbol{w}) - \mathcal{L}(\boldsymbol{w})] \\ &\quad = \Delta\boldsymbol{w}^T g^{\boldsymbol{w}} + \frac{1}{2}\Delta\boldsymbol{w}^T \boldsymbol{H}^{\boldsymbol{w}}\Delta\boldsymbol{w} + O(\|\Delta\boldsymbol{w}\|^3) \\ &\quad \approx \frac{1}{2}\Delta\boldsymbol{w}^T \boldsymbol{H}^{\boldsymbol{w}}\Delta\boldsymbol{w}\end{aligned}$$

$$(5)$$

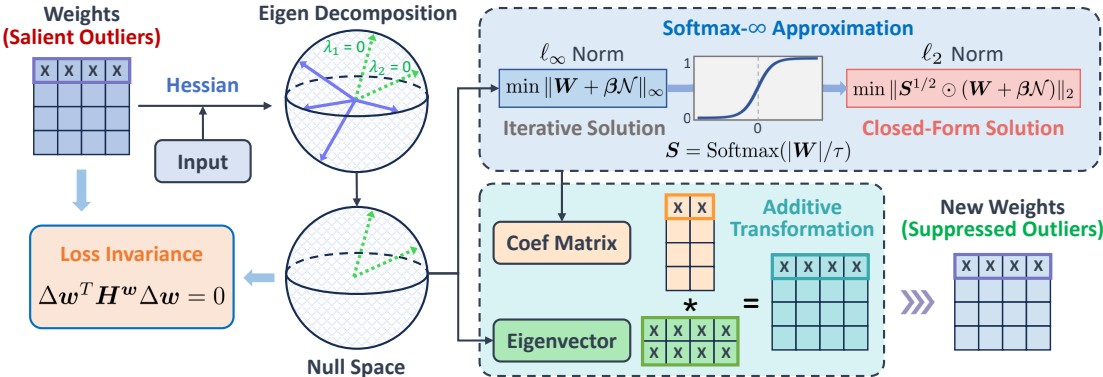

Figure 3. Workflow of the proposed OSAQ. Under the condition of loss invariance, we explicitly solve for a coefficient matrix to weight the vectors in the Hessian null space, thereby constructing an additive transformation that effectively suppresses outliers in weights.

where $\boldsymbol{w}$ is the flattened vector of weights $\boldsymbol{W}$, $\boldsymbol{g^w} = \mathbb{E}[\nabla_{\boldsymbol{w}}\mathcal{L}(\boldsymbol{w})]$ is the first-order gradient, $\boldsymbol{H^w} = \mathbb{E}[\nabla_{\boldsymbol{w}}^2\mathcal{L}(\boldsymbol{w})]$ is the second-order Hessian.

As illustrated in Figure 1, we observe that although the low-rank structures of different inputs themselves are inconsistent, the Hessian $\boldsymbol{H^w}$ exhibits a pronounced low-rank property. Specifically, along certain feature directions, the corresponding eigenvalues vanish, indicating that their magnitudes are effectively zero. The associated eigenvectors collectively form the null space of $\boldsymbol{H^w}$. More importantly, this null space remains stable across different input samples, meaning that the eigenvectors within it do not change. This reveals the low-rank consistency of Hessian.

**Insights: Loss-Invariant Additive Transformation.** By definition of the null space (Strang, 2022), multiplying $\boldsymbol{H^w}$ by any vector within the null space yields zero. Therefore, by forming a weighted combination of these null-space vectors, we can construct $\Delta\boldsymbol{W}$. This enables an additive transformation that guarantees the loss remains unchanged:

$$\boldsymbol{W'} = \boldsymbol{W} + \Delta\boldsymbol{W} \quad \text{s.t. } \Delta\boldsymbol{w}^T \boldsymbol{H^w} \Delta\boldsymbol{w} = 0 \qquad (6)$$

where $\Delta\boldsymbol{w}$ is the flattened vector. In this way, although the transformation is not equivalent as in the multiplicative cases, it can still ensure that model performance remains unaffected. Furthermore, with careful optimization, if $\Delta\boldsymbol{W}$ can counteract the influence of outliers, it will serve as a promising strategy for outlier suppression. Since this transformation operates solely on the weights themselves, without adjustment or compensation from the other layers, it is referred to as outlier self-absorption. Figure 3 illustrates the overall workflow of this approach.

## 5. Outlier Self-Absorption Quantization

Building on the above observations and insights, we aim to construct a low-rank-guided $\Delta\boldsymbol{W}$ to perform an addi-

tive transformation on the weights, enabling outlier self-absorption while preserving model performance. Given a weight matrix $\boldsymbol{W} \in \mathbb{R}^{M \times N}$, with $M$ as the output channel dimension and $N$ as the input channel dimension, the construction process of $\Delta\boldsymbol{W}$ is detailed below.

**Extraction of Null Space.** First, we perform eigen-decomposition on the Hessian $\boldsymbol{H^w}$ and arrange its eigenvalues in non-decreasing order by magnitude, as follows:

$$\boldsymbol{H^w} = \boldsymbol{V} \operatorname{diag}\left(\lambda_1, \ldots, \lambda_N\right) \boldsymbol{V}^T, \ \ 0 \le |\lambda_1| \le \cdots \le |\lambda_N| \qquad (7)$$

where $\boldsymbol{V} \in \mathbb{R}^{N \times N}$ is the matrix of eigenvectors, and $\lambda_1, \ldots, \lambda_N$ are the corresponding eigenvalues. Typically, the eigenvalues $\lambda$ are not exactly zero. Using a fixed threshold directly may lead to imbalanced null space dimensions across different layers. Therefore, we adopt a tail-energy strategy, where we start from the smallest eigenvalues and accumulate them to obtain the prefix energy, and the null-space dimension is determined as the smallest $K$ such that the cumulative tail energy reaches a predefined threshold, as follows:

$$\mathcal{N} = \boldsymbol{V}_{[:,0:K-1]}^T, \text{ where } K = \min_k\left\{\sum_{i=1}^k |\lambda_i| \ge \gamma \sum_{i=1}^N |\lambda_i|\right\} \qquad (8)$$

In this equation, $\gamma \in (0, 1)$ is the tail-energy threshold. $\mathcal{N} \in \mathbb{R}^{K \times N}$ denotes the null space of $\boldsymbol{H^w}$, where each row corresponds to a feature direction along which $\boldsymbol{H^w}$ exhibits vanishing curvature.

**Softmax-∞ Objective Approximation.** After obtaining the null space, we introduce a weighting coefficient matrix $\boldsymbol{\beta} \in \mathbb{R}^{M \times K}$ to assign weights to each vector within the null space, thereby constructing $\Delta\boldsymbol{W}$, i.e., $\Delta\boldsymbol{W} = \boldsymbol{\beta}\mathcal{N}$. Our objective is to minimize the numerical range of the weights after applying the additive perturbation. A straightforward

approach is to minimize the $\ell_\infty$ norm as follows:

$$\min_{\boldsymbol{\beta}} \|\boldsymbol{W} + \Delta\boldsymbol{W}\|_\infty = \min_{\boldsymbol{\beta}} \|\boldsymbol{W} + \boldsymbol{\beta}\mathcal{N}\|_\infty \qquad (9)$$

However, the $\ell_\infty$ norm is non-differentiable, which requires iterative optimization. To address this, we adopt a Softmax-$\infty$ approximation (Boyd & Vandenberghe, 2004), which approximates the original $\ell_\infty$ norm by optimizing the differentiable $\ell_2$ norm of the softmax-scaled values. Specifically, we apply the softmax operation along the output-channel dimension, as follows:

$$s_{ij} = \frac{\exp\left(|W_{ij}|/\tau\right)}{\sum_{t=1}^{N} \exp\left(|W_{it}|/\tau\right)} \qquad (10)$$

where $i = 1, \cdots, M$, and $\tau > 0$ is a temperature coefficient. When $\tau$ is large, it captures the average behavior across all components, while as $\tau \to 0^+$, it increasingly emphasizes extreme peak values. In this case, applying the $\ell_2$ norm to the peak-emphasized parameters can serve as an approximation of the $\ell_\infty$ norm, thereby enabling effective identification and suppression of outliers.

**Explicit Solution of Coefficient Matrix $\boldsymbol{\beta}$.** Next, we formulate the optimization objective as a softmax-scaled $\ell_2$ norm. Since the quantization scale and zero-point are both computed along the output-channel dimension, for notational clarity, we explicitly present the $\ell_2$ norm optimization objective for each output channel. The objective for the $i$-th output channel is given as follows:

$$\min_{\boldsymbol{b}_i} \frac{1}{2} \sum_{j=1}^{N} s_{ij} \left(W_{ij} + \boldsymbol{b}_i^T \boldsymbol{n}_j\right)^2 + \frac{\mu_1}{2} \|\boldsymbol{b}_i\|_2^2 + \frac{\mu_2}{2} \left(\boldsymbol{b}_i^T \boldsymbol{v}\right)^2 \qquad (11)$$

where $\boldsymbol{b}_i = \boldsymbol{\beta}[i,:] \in \mathbb{R}^K$, $\boldsymbol{n}_j = \mathcal{N}[:,j] \in \mathbb{R}^K$, $\boldsymbol{v} = \mathcal{N}\mathbf{1}_N \in \mathbb{R}^K$, and $\mu_1, \mu_2 > 0$ are the balancing coefficients. In the above optimization objective, the first term is the $\ell_2$ norm, which serves to reduce the numerical range and suppress outliers within each channel; the second term is a regularization term on $\boldsymbol{b}_i$, intended to prevent excessive corrections; the third term imposes a anti-shift constraint, which penalizes uniform translations of the entire channel in the same direction.

By taking the derivative of the objective function with respect to $\boldsymbol{b}_i$ and setting the first-order optimality condition to zero, we obtain the normal equation as follows:

$$
\begin{aligned}
\boldsymbol{A}_i \boldsymbol{b}_i &= -\boldsymbol{\rho}_i, \\
\boldsymbol{A}_i &= \sum_{j=1}^{N} s_{ij} \boldsymbol{n}_j \boldsymbol{n}_j^T + \mu_1 \boldsymbol{I}_K + \mu_2 \boldsymbol{v}\boldsymbol{v}^T, \\
\boldsymbol{\rho}_i &= \sum_{j=1}^{N} s_{ij} W_{ij} \boldsymbol{n}_j
\end{aligned} \qquad (12)
$$

Thus, we can obtain the closed-form optimal solution of $\boldsymbol{b}_i$ under the first-order optimality condition, and by combining all $\boldsymbol{b}_i$, we finally construct the complete coefficient matrix $\boldsymbol{\beta}$ as follows:

$$
\begin{aligned}
\boldsymbol{\beta}^* &= [\boldsymbol{b}_1^*, \cdots, \boldsymbol{b}_M^*]^T, \\
\text{where } \boldsymbol{b}_i^* &= -\boldsymbol{A}_i^{-1}\boldsymbol{\rho}_i, \ i = 1, \cdots, M
\end{aligned} \qquad (13)
$$

**Remark 1.** Each rank-one matrices $\boldsymbol{n}_j \boldsymbol{n}_j^T$ and $\boldsymbol{v}\boldsymbol{v}^T$ is positive semi-definite, and $\mu_1 \boldsymbol{I}_K$ is strictly positive definite. Since $s_{ij} > 0$, $\mu_1 > 0$, and $\mu_2 > 0$, it follows that $\boldsymbol{A}_i$, the second-order derivative of the objective function with respect to $\boldsymbol{b}_i$, is symmetric positive definite and thus invertible, i.e., $\boldsymbol{A}_i \succeq \mu_1 \boldsymbol{I}_K \succ \boldsymbol{0}$. Consequently, $\boldsymbol{b}_i^*$ exists, is unique, and is the unique global minimizer.

# 6. Experiments

## 6.1. Experimental Setups

**Models and Datasets.** We perform quantization on popular pre-trained LLMs, including LLaMA2 (7B, 13B, 70B) (Touvron et al., 2023), and LLaMA3 (8B, 70B) (Dubey et al., 2024), and larger-scale instruction-tuned LLMs, including Mistral-Large-123B-Instruct (Mistral AI Team, 2024) and Llama-3.1-405B-Instruct (Dubey et al., 2024). We evaluate the performance on language generation tasks using perplexity on WikiText2 (Merity et al., 2016) and C4 (Raffel et al., 2020) datasets, and on commonsense QA tasks using zero-shot accuracy on PIQA (Bisk et al., 2020), ARC (Clark et al., 2018), and WinoGrande (Sakaguchi et al., 2021) datasets. We also assess the models on MMLU (Hendrycks et al., 2020) and MT-Bench (Zheng et al., 2023) benchmarks.

**Baselines.** We compare the proposed OSAQ with strong weight-only quantization baselines, including GPTQ (Frantar et al., 2022), AWQ (Lin et al., 2023), QuIP (Chee et al., 2023), MagR (Zhang et al., 2024), and OmniQuant (Shao et al., 2023). We also consider quantizing both the weights and KV-Cache, and compare with WKVQuant (Yue et al., 2024). In the 2-bit quantization setting, we further enhance performance by incorporating coordinate descent iterations (Behdin et al., 2023), denoted by the † symbol.

**Implementation Details.** We primarily focus on weight-only quantization, while also exploring KV-Cache quantization. The proposed QSAQ serves as a plug-and-play component that complements existing methods to further enhance performance. Accordingly, the quantization setup is aligned with the respective baselines. For example, when combined with GPTQ, the calibration data consists of 128 samples with a sequence length of 2048. For the selection of hyperparameters, we perform a simple grid search as in Figure 5, which demonstrates that the quantization performance is robust to the choice of these values.

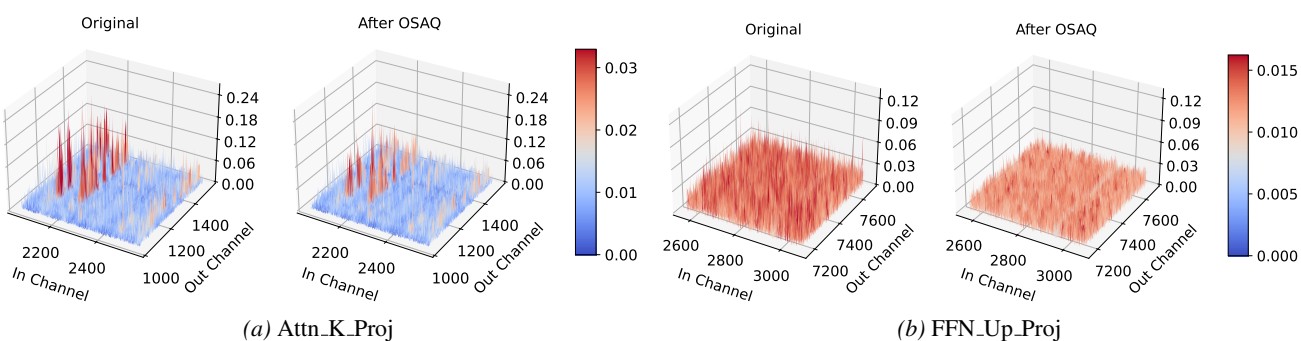

*(a)* Attn_K_Proj                                    *(b)* FFN_Up_Proj

*Figure 4.* Weight distributions before and after additive transformation in LLaMA2-7B's 0-th layer. The original model exhibits prominent outliers, whereas OSAQ suppresses them, substantially reducing quantization difficulty. More results are provided in Appendix A.

*Table 1.* Perplexity results (↓) of quantized models on language generation tasks. This table reports the results of LLaMA2 and LLaMA3 models of various scales under different quantization settings.

| Prec. | Method | WikiText2 | | | | | C4 | | | | |
|---|---|---|---|---|---|---|---|---|---|---|---|
| | | 2-7B | 2-13B | 2-70B | 3-8B | 3-70B | 2-7B | 2-13B | 2-70B | 3-8B | 3-70B |
| FP16 | Baseline | 5.47 | 4.88 | 3.31 | 6.10 | 2.90 | 6.97 | 6.46 | 5.52 | 9.20 | 6.90 |
| W4A16 | RTN | 6.11 | 5.20 | 3.67 | 8.70 | - | 7.71 | 6.83 | 5.79 | 14.0 | - |
| | MagR | 5.91 | 5.17 | 3.58 | - | - | 7.52 | 6.81 | 5.72 | - | - |
| | OmniQuant | 5.74 | 5.02 | 3.47 | - | - | 7.35 | 6.65 | 5.65 | - | - |
| | QuIP | 5.94 | 5.01 | 3.53 | 6.50 | 3.40 | 8.01 | 6.88 | 5.87 | 11.1 | 7.10 |
| | AWQ | 6.15 | 5.12 | 3.57 | 7.10 | 3.70 | 7.68 | 6.74 | 5.71 | 10.1 | 7.40 |
| | OSAQ+AWQ | **5.99** | **5.04** | **3.53** | **6.82** | **3.57** | **7.50** | **6.67** | **5.66** | **9.93** | **7.22** |
| | GPTQ | 5.83 | 5.13 | 3.58 | 7.00 | 3.60 | 7.37 | 6.70 | 5.67 | 11.8 | 7.40 |
| | OSAQ+GPTQ | **5.73** | **5.04** | **3.48** | **6.82** | **3.42** | **7.34** | **6.64** | **5.61** | **11.5** | **7.24** |
| W3A16 g128 | RTN | 6.66 | 5.51 | 3.97 | 27.9 | 11.8 | 8.40 | 7.18 | 6.02 | 110 | 22.0 |
| | MagR | 6.46 | 5.45 | 3.95 | - | - | 8.22 | 7.12 | 6.00 | - | - |
| | OmniQuant | 6.03 | 5.28 | 3.78 | - | - | 7.75 | 6.98 | 5.85 | - | - |
| | AWQ | 6.24 | 5.32 | 3.83 | 8.20 | 4.80 | 7.84 | 6.94 | 5.82 | 11.6 | 8.00 |
| | OSAQ+AWQ | **6.08** | **5.23** | **3.75** | **7.96** | **4.61** | **7.75** | **6.89** | **5.72** | **11.4** | **7.87** |
| | GPTQ | 6.29 | 5.42 | 3.85 | 8.20 | 5.20 | 7.89 | 7.00 | 5.85 | 13.7 | 10.5 |
| | OSAQ+GPTQ | **6.07** | **5.30** | **3.79** | **7.98** | **4.99** | **7.77** | **6.95** | **5.78** | **13.4** | **10.2** |
| W3A16 | RTN | 539 | 10.7 | 7.52 | 2.2e3 | 3.2e4 | 402 | 12.5 | 10.0 | 560 | 112 |
| | MagR | 8.66 | 6.55 | 4.64 | - | - | 10.8 | 8.26 | 6.77 | - | - |
| | AWQ | 24.0 | 10.5 | 6.29 | 12.8 | 5.92 | 23.9 | 13.1 | 7.11 | 16.8 | 9.71 |
| | OmniQuant | 6.58 | 5.58 | 3.92 | - | - | 8.65 | 7.44 | 6.06 | - | - |
| | GPTQ | 8.37 | 6.44 | 4.82 | 13.0 | 5.88 | 9.81 | 8.02 | 6.57 | 45.9 | 9.66 |
| | OSAQ+GPTQ | **6.75** | **5.72** | **4.21** | **11.3** | **5.49** | **8.70** | **7.54** | **6.09** | **23.1** | **8.62** |
| | QuIP | 6.50 | 5.34 | 3.85 | 7.50 | 4.70 | 8.74 | 7.34 | 6.14 | 11.3 | 8.00 |
| | OSAQ+QuIP | **6.37** | **5.25** | **3.81** | **7.43** | **4.65** | **8.68** | **7.27** | **6.06** | **10.8** | **7.83** |
| W2A16 g128 | RTN | 1.2e4 | 5.8e3 | 7.8e3 | 1.9e3 | 4.6e5 | 7.2e4 | 3.9e3 | 3.6e3 | 2.5e4 | 4.7e5 |
| | MagR† | 9.94 | 7.63 | 5.52 | - | - | 14.1 | 10.6 | 8.05 | - | - |
| | OmniQuant | 11.1 | 8.26 | 6.55 | - | - | 15.0 | 11.1 | 8.52 | - | - |
| | GPTQ | 36.8 | 28.1 | 19.2 | 210 | 11.9 | 33.7 | 21.0 | 13.8 | 4.1e4 | 22.8 |
| | OSAQ+GPTQ | **21.2** | **13.4** | **10.7** | **63.8** | **8.36** | **18.3** | **13.8** | **9.44** | **352** | **14.7** |
| | OSAQ+GPTQ† | **10.6** | **7.60** | **5.97** | **26.1** | **6.11** | **14.7** | **10.5** | **8.42** | **62.0** | **10.2** |

## 6.2. Main Results

**Visualization of weight distributions.** Figure 4 illustrates the weight distributions before and after applying the additive transformation. After applying OSAQ, the outliers are effectively suppressed, resulting in a more compact distribution, which substantially reduces quantization difficulty.

**Evaluation on Language Generation Tasks.** Table 1 reports perplexity results of LLaMA2 and LLaMA3 models.

*Table 2.* Zero-shot accuracy (↑) of quantized models on commonsense QA tasks. This table reports the results of LLaMA3 models of various scales under different quantization settings.

| Prec. | Method | LLaMA3-8B | | | | | LLaMA3-70B | | | | |
|---|---|---|---|---|---|---|---|---|---|---|---|
| | | PIQA | ARC-e | ARC-c | Wino | Avg. | PIQA | ARC-e | ARC-c | Wino | Avg. |
| FP16 | Baseline | 79.9 | 80.1 | 50.4 | 72.8 | 70.8 | 82.4 | 86.9 | 60.3 | 80.6 | 77.6 |
| W4A16 | QuIP | 78.2 | 78.2 | 47.4 | **73.2** | 69.2 | **82.5** | 86.0 | 58.7 | 79.7 | 76.7 |
| | OSAQ$_{+QuIP}$ | **78.8** | **78.9** | **48.0** | 73.1 | **69.7** | 82.4 | **86.3** | **60.0** | **80.1** | **77.2** |
| W3A16 g128 | AWQ | 77.7 | 74.0 | 43.2 | 72.1 | 66.8 | 81.4 | 84.7 | 58.0 | 78.6 | 75.7 |
| | OSAQ$_{+AWQ}$ | **78.2** | **75.1** | **43.9** | **72.4** | **67.4** | **82.1** | **85.4** | **58.9** | **79.2** | **76.4** |
| | GPTQ | 74.9 | 70.5 | 37.7 | 71.1 | 63.6 | 80.6 | 79.6 | 52.1 | 77.1 | 72.4 |
| | OSAQ$_{+GPTQ}$ | **76.4** | **72.2** | **39.7** | **72.3** | **65.2** | **81.9** | **82.3** | **54.7** | **78.7** | **74.4** |
| W3A16 | QuIP | 76.8 | 72.9 | 41.0 | 72.5 | 65.8 | 82.3 | 83.3 | 54.9 | 78.4 | 74.7 |
| | OSAQ$_{+QuIP}$ | **77.6** | **73.5** | **41.6** | **72.7** | **66.4** | 82.3 | **84.0** | **55.5** | **78.9** | **75.2** |
| W2A16 g128 | GPTQ | 53.9 | 28.8 | 19.9 | 50.5 | 38.3 | 62.7 | 38.9 | 24.6 | 59.9 | 46.5 |
| | OSAQ$_{+GPTQ}$ | 58.1 | 38.6 | 25.8 | 54.9 | 44.4 | 65.5 | 42.5 | 32.0 | 63.8 | 51.0 |
| | OSAQ$_{+GPTQ†}$ | **64.6** | **44.2** | **37.7** | **60.4** | **51.7** | **69.2** | **49.0** | **39.7** | **68.8** | **56.7** |

*Table 3.* Zero-shot accuracy (↑) of quantized models on MMLU benchmark. This table reports the results of LLaMA2-7B and LLaMA3-8B models under different quantization settings.

| Prec. | Method | LLaMA2-7B | | | | | LLaMA3-8B | | | | |
|---|---|---|---|---|---|---|---|---|---|---|---|
| | | STEM | Hums | Social | Others | Avg. | STEM | Hums | Social | Others | Avg. |
| FP16 | Baseline | 34.4 | 39.8 | 47.3 | 47.1 | 42.2 | 53.8 | 54.9 | 73.3 | 70.4 | 63.1 |
| W4A16 | OmniQuant | 28.8 | 32.2 | 34.7 | 35.8 | 32.9 | 49.4 | 49.1 | 66.6 | 64.4 | 57.4 |
| | GPTQ | 32.7 | 36.9 | 42.6 | 42.6 | 38.7 | 47.3 | 52.3 | 66.0 | 64.9 | 57.6 |
| | OSAQ$_{+GPTQ}$ | **33.6** | **37.5** | **43.5** | **43.9** | **39.6** | **48.0** | **52.7** | **66.7** | **65.3** | **58.2** |
| W3A16 | OmniQuant | 29.1 | 31.1 | 30.6 | 30.4 | 30.3 | 26.3 | 27.8 | 29.5 | 29.9 | 28.4 |
| | GPTQ | 28.2 | 27.0 | 32.1 | 29.9 | 29.3 | 26.2 | 29.2 | 34.4 | 30.0 | 30.0 |
| | OSAQ$_{+GPTQ}$ | **29.3** | **29.4** | **33.8** | **31.1** | **30.9** | **27.0** | **30.3** | **35.3** | **31.1** | **30.9** |

*Table 4.* Quantization results of larger-scale instruction-tuned models, including models of 123B and 405B. This table reports zero-shot accuracy (↑) on ARC and MMLU benchmarks.

| Model. | Prec. | Method | ARC-e | ARC-c | MMLU | | | | |
|---|---|---|---|---|---|---|---|---|---|
| | | | | | STEM | Hums | Social | Others | Avg. |
| Mistral-Large-123B-Instruct | W4A16 | LeanQuant | 85.1 | 64.0 | 76.6 | 77.3 | 89.2 | 85.9 | 82.3 |
| | | GPTQ | 84.6 | 64.0 | 76.3 | 77.2 | 89.3 | 85.2 | 82.0 |
| | | OSAQ$_{+GPTQ}$ | **85.0** | **64.1** | **76.7** | **77.4** | **89.3** | **85.7** | **82.3** |
| Llama-3.1-405B-Instruct | W4A16 g128 | LeanQuant | 88.3 | 64.8 | 82.7 | 83.2 | 90.6 | 87.7 | 86.1 |
| | | GPTQ | 88.2 | 65.1 | 82.3 | 82.6 | 90.5 | 87.5 | 85.7 |
| | | OSAQ$_{+GPTQ}$ | **88.3** | **65.0** | **82.6** | **83.2** | **90.8** | **87.7** | **86.1** |

In the W3A16 setting, OSAQ+GPTQ reduces perplexity on LLaMA2-13B from 6.44 to 5.72.

**Evaluation on Commonsense QA Tasks.** Table 2 shows that OSAQ consistently improves zero-shot accuracy of LLaMA3 models. In the 4-bit setting, OSAQ brings stable gains over QuIP, maintaining accuracy close to FP16 baseline. In 3-bit quantization on LLaMA3-8B, OSAQ+GPTQ increases the average accuracy from 63.6% to 65.2%.

**Evaluation on MMLU Benchmark.** Table 3 reports the

zero-shot accuracy on the MMLU benchmark for LLaMA2-7B and LLaMA3-8B. OSAQ+GPTQ raises the average score from 38.7% to 39.6% on LLaMA2-7B and from 57.6% to 58.2% on LLaMA3-8B in the 4-bit setting.

**Evaluation of Larger Instruction-Tuned Models.** Table 4 presents quantization results for instruction-tuned models with 123B and 405B parameters. The results show that even at this scale, OSAQ remains effective. On LLaMA-3.1-405B-Instruct, OSAQ+GPTQ attains an average of 86.1% on MMLU, outperforming vanilla GPTQ.

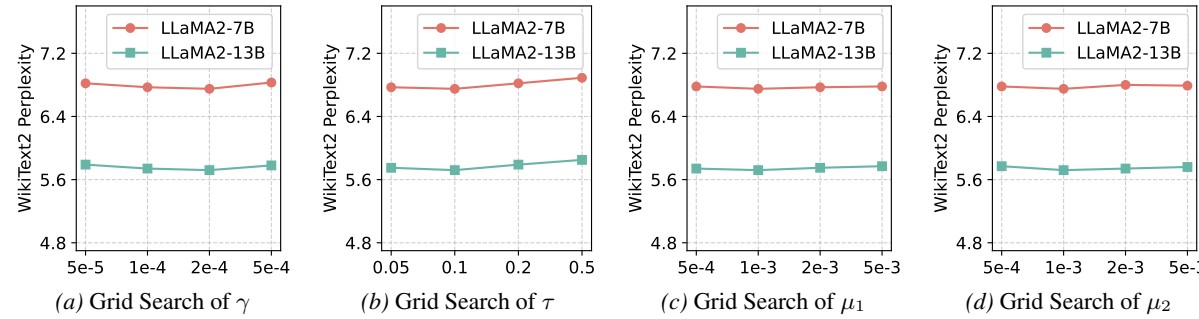

*(a) Grid Search of $\gamma$*   *(b) Grid Search of $\tau$*   *(c) Grid Search of $\mu_1$*   *(d) Grid Search of $\mu_2$*

*Figure 5.* Grid search of hyperparameters for LLaMA2 models under the 3-bit quantization setting. The results show that the quantization performance is robust to the choice of hyperparameters.

*Table 5.* Impact evaluation of additive transformation on original FP model performance.

| Prec. | Method | WikiText2 | | C4 | |
|---|---|---|---|---|---|
| | | 2-7B | 2-13B | 2-7B | 2-13B |
| FP16 | Baseline | 5.47 | 4.88 | 6.97 | 6.46 |
| | Baseline+Add. | 5.52 | 4.95 | 7.01 | 6.54 |
| W3A16 | GPTQ | 8.37 | 6.44 | 9.81 | 8.02 |
| | GPTQ+Add. | **6.75** | **5.72** | **8.70** | **7.54** |

*Table 6.* Evaluation of null space stability via maximum singular value of $\mathcal{N}_1^\top \mathcal{N}_2$ across different layers and modules.

| Layer | Layer 0 | | | Layer 7 | | |
|---|---|---|---|---|---|---|
| | k_proj | q_proj | v_proj | k_proj | q_proj | v_proj |
| Max singular value of $\mathcal{N}_1^\top \mathcal{N}_2$ | 0.973 | 0.981 | 0.975 | 0.966 | 0.970 | 0.965 |

*Table 7.* Effect of Softmax-$\infty$ approximation compared to directly applying $\ell_2$ norm.

| Prec. | Method | WikiText2 | | C4 | |
|---|---|---|---|---|---|
| | | 2-7B | 2-13B | 2-7B | 2-13B |
| FP16 | Baseline | 5.47 | 4.88 | 6.97 | 6.46 |
| W3A16 | $\ell_2$ norm | 7.82 | 6.11 | 9.13 | 7.88 |
| | Softmax-$\infty$+$\ell_2$ norm | **6.75** | **5.72** | **8.70** | **7.54** |

**Evaluation with Quantized Activations.** Since OSAQ offers excellent compatibility, we also perform evaluation on KV-Cache quantization and weight-activation quantization, as shown in Appendix B. In this case, applying OSAQ to suppress weight outliers is still beneficial, as it not only mitigates the outliers themselves but also facilitates subsequent weight–activation transformations.

### 6.3. Ablation Studies

**Impact of Additive Transformation on FP Model.** Table 5 shows that additive transformation has only a negligible impact on FP16 models (e.g., from 5.47 to 5.52 on WikiText2 for LLaMA2-7B). For instance, GPTQ+Add. reduces perplexity from 8.37 to 6.75 on WikiText2. This confirms that the additive transformation preserves FP performance while significantly improving low-bit quantization.

**Stability of Null Space.** We quantitatively analyze the alignment between the null spaces $\mathcal{N}_1$ and $\mathcal{N}_2$ computed from two different input batches across multiple layers, as shown in Table 6. The singular values of $\mathcal{N}_1^\top \mathcal{N}_2$ correspond to the cosines of the principal angles between the two subspaces, and values closer to 1 indicate stronger alignment. More experiments, including studies on data distribution and calibration set size, are provided in Appendix D.

**Effect of Softmax-$\infty$ Approximation.** Table 7 compares Softmax-$\infty$ approximation with directly applying $\ell_2$ norm. Direct $\ell_2$ norm optimization is less effective in suppressing outliers, leading to higher perplexity. In contrast, Softmax-$\infty$ approximation, which serves as a differentiable proxy of

the $\ell_\infty$ norm, achieves substantially lower perplexity.

**Selection of Hyperparameters.** Figure 5 shows grid search results of the hyperparameters. $\gamma$ controls null space size, $\tau$ adjusts outlier emphasis in the Softmax-$\infty$ approximation, $\mu_1$ regulates additive strength, and $\mu_2$ suppresses bias shifts. The results remain stable across different values, demonstrating robustness to hyperparameter choice.

## 7. Conclusion

In this work, we introduced OSAQ, an outlier self-absorption method based on additive transformation for low-bit weight-only quantization of LLMs. By exploiting the second-order low-rank property of the Hessian, OSAQ identifies a stable null space and constructs an additive transformation that effectively suppresses weight outliers while preserving task loss. The additive transformation is fully absorbed into the weights, requiring no adjustment of neighboring layers and incurring no additional inference cost. Furthermore, the construction coefficients are obtained in closed form, making it free from training or iterative optimization. Extensive experiments across multiple models and tasks demonstrate the effectiveness of OSAQ.

## Acknowledgements

This work was supported in part by the Strategic Priority Research Program of Chinese Academy of Sciences under Grant Number XDB1100000; in part by the National Natural Science Foundation of China under Grant Number 62276255; in part by the Postdoctoral Fellowship Program of CPSF under Grant Number GZC20251175.

## Impact Statement

This paper presents work whose goal is to advance the field of Machine Learning. There are many potential societal consequences of our work, none which we feel must be specifically highlighted here.

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

# A. More Visualization Results of Weight Distributions

Here, we provide additional visualization results to further illustrate the effect of the proposed method. Figure 6 presents the weight distributions in the attention module of the 0-th layer of LLaMA2-7B, inluding k_proj, v_proj, q_proj, and o_proj, while Figure 7 shows the corresponding results in the MLP module, including up_proj, gate_proj, and down_proj.

In both cases, the original model exhibits prominent outliers that significantly enlarge the weight range, which in turn increases the difficulty of representing weights under low-bit quantization. By approximating the $\ell_\infty$ norm in the coefficient optimization, OSAQ achieves strong suppression of these outliers, effectively tightening the distribution and reducing the numerical range. As a result, the overall weight distribution becomes smoother and more regular, making the quantization process more stable and yielding better quantization performance.

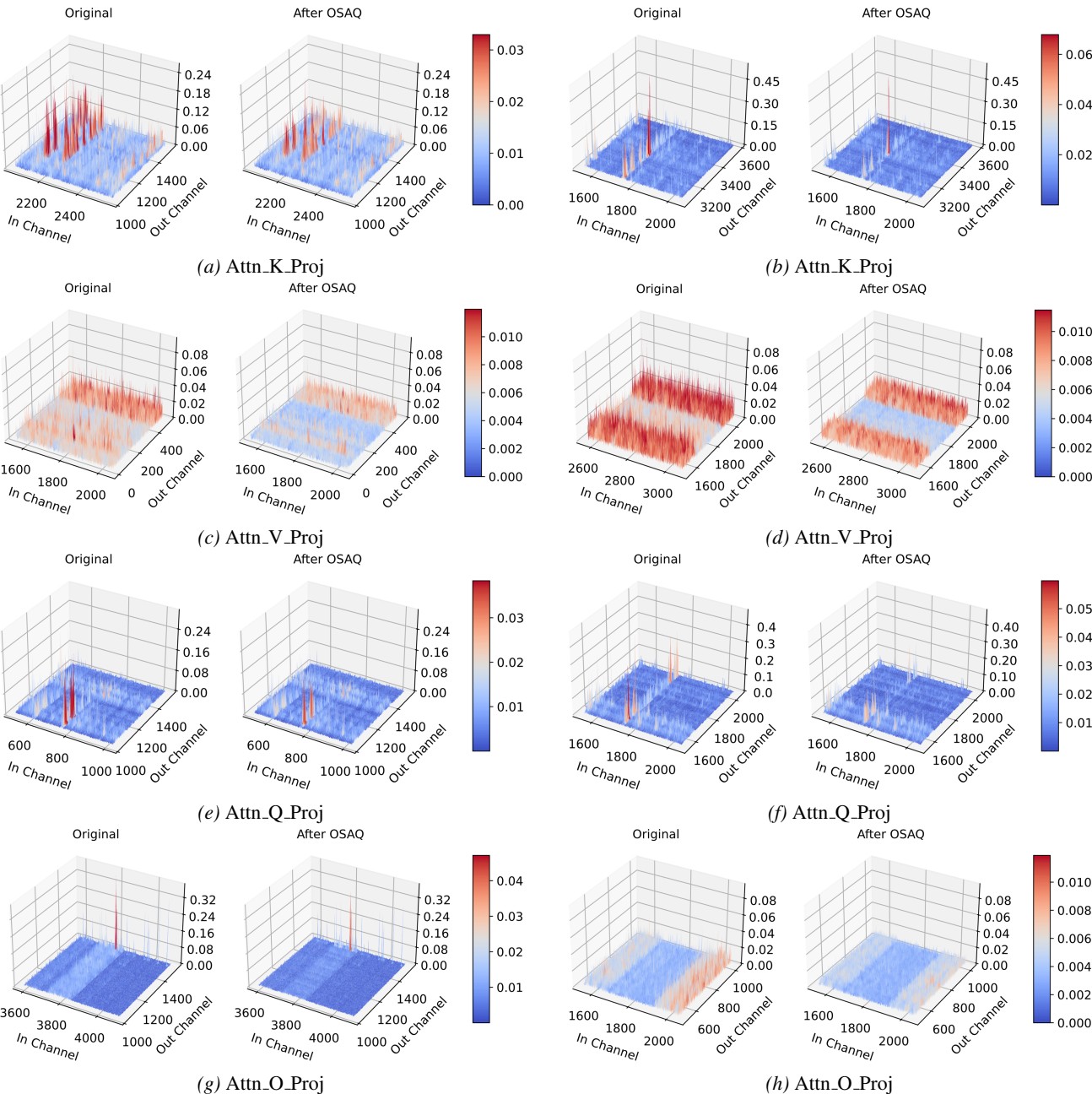

*Figure 6.* Weight distributions before and after the additive transformation in 0-th layer's attention module of LLaMA2-7B. The original model exhibits prominent outliers, whereas OSAQ effectively suppresses them, substantially reducing the difficulty of quantization.

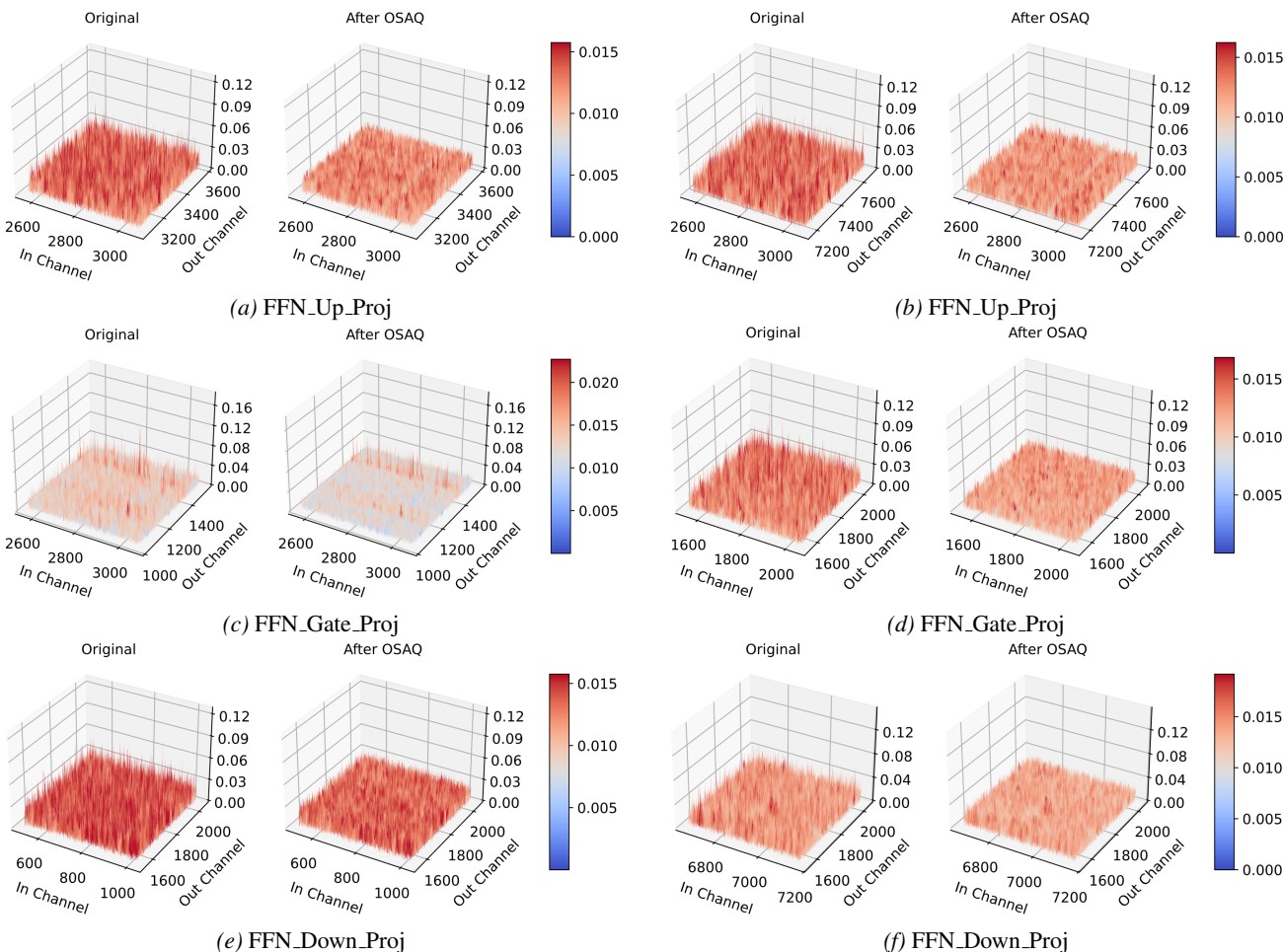

*Figure 7.* Weight distributions before and after the additive transformation in 0-th layer's MLP module of LLaMA2-7B. The original model exhibits prominent outliers, whereas OSAQ effectively suppresses them, substantially reducing the difficulty of quantization.

## B. Evaluation of Weight-Activtion Quantization

OSAQ applies an additive perturbation to the weights to mitigate outliers, and its primary goal is to address issues arising in weight-only quantization. Accordingly, in our main experimental comparisons, we evaluate OSAQ against several widely used weight-only methods, including GPTQ, AWQ, and QuIP.

In our implementation, OSAQ is used as a plug-and-play component, and when combined with other methods, it is always applied before them. OSAQ first suppresses weight outliers, making the weights easier to quantize, and then the subsequent quantization method is applied. Therefore, OSAQ is fully compatible with all compared approaches.

Therefore, we also conducted experiments on weight-activation quantization, including experiments that quantize only the KV-cache activations as well as experiments that quantize all activations.

**Evaluation of KV-Cache Quantization.** Table 8 reports perplexity results when both weights and KV-Cache are quantized to 4-bits, which validates OSAQ in another practical configuration that also quantizes the KV-Cache. For LLaMA2-7B, perplexity on WikiText2 decreases from 5.64 to 5.59 and on C4 from 7.49 to 7.43, while for LLaMA2-13B it decreases from 5.00 to 4.95 on WikiText2 and from 6.89 to 6.81 on C4.

**Evaluation of Weight-Activtion Quantization.** In weight-activation quantization, outliers in the activation distribution typically become the dominant performance bottleneck, which is why existing studies place greater emphasis on activation handling. Representative methods such as QuaRot (Ashkboos et al., 2024), DuQuant (Lin et al., 2024), and SpinQuant (Liu et al., 2024) are all designed specifically to address activation-side challenges. Despite this, we additionally conduct

*Table 8.* Quantization results under the 4-bit weights and 4-bit KV-Cache setting. This table reports perplexity results (↓) of LLaMA2 models of different scales.

| Prec. | Method | WikiText2 | | C4 | |
|---|---|---|---|---|---|
| | | 2-7B | 2-13B | 2-7B | 2-13B |
| FP16 | Baseline | 5.47 | 4.88 | 6.97 | 6.46 |
| W4A16KV4 | OmniQuant | 6.09 | 5.18 | 8.98 | 7.30 |
| | WKVQuant | 5.64 | 5.00 | 7.49 | 6.89 |
| | OSAQ$_{+WKVQuant}$ | **5.59** | **4.95** | **7.43** | **6.81** |

*Table 9.* Quantization results under the 4-bit weights and 4-bit activations setting. This table reports perplexity results (↓) of LLaMA2 models of different scales.

| Prec. | Method | WikiText2 | | C4 | |
|---|---|---|---|---|---|
| | | 2-7B | 2-13B | 2-7B | 2-13B |
| FP16 | Baseline | 5.47 | 4.88 | 6.97 | 6.46 |
| W4A4 | QuaRot | 6.10 | 5.40 | 7.82 | 6.97 |
| | OSAQ$_{+QuaRot}$ | **6.03** | **5.32** | **7.73** | **6.85** |
| | DuQuant | 6.28 | 5.42 | 7.90 | 7.05 |
| | OSAQ$_{+DuQuant}$ | **6.19** | **5.34** | **7.78** | **6.96** |
| | SpinQuant | 5.90 | 5.30 | 7.79 | 6.93 |
| | OSAQ$_{+SpinQuant}$ | **5.84** | **5.25** | **7.71** | **6.86** |

experiments on weight-activation quantization, and the results on WikiText2 are summarized in Table 9.

As we can see, applying OSAQ to suppress weight outliers in weight-activation quantization still yields benefits. On the one hand, the weight outliers themselves are mitigated; on the other hand, this preprocessing step can also implicitly facilitate the downstream weight-activation transformations performed by other methods.

## C. Evaluation on MT-Bench Benchmark

MT-Bench is a multi-turn dialogue benchmark designed to evaluate the instruction-following and reasoning abilities of LLMs. Table 10 reports the MT-Bench performance of LLaMA2 models under different quantization settings. Compared with the GPTQ baseline, OSAQ consistently improves the performance for both 7B and 13B models under W4A16 and W3A16 settings, demonstrating its effectiveness in mitigating the performance degradation caused by low-bit weight quantization.

*Table 10.* Quantization results on MT-Bench benchmark. This table reports MT-Bench scores (↑) of LLaMA2 models of different scales.

| Prec. | Method | MT-Bench | |
|---|---|---|---|
| | | 2-7B | 2-13B |
| FP16 | Baseline | 3.83 | 4.69 |
| W4A16 | GPTQ | 3.66 | 4.50 |
| | OSAQ$_{+GPTQ}$ | **3.72** | **4.55** |
| W3A16 | GPTQ | 3.37 | 4.32 |
| | OSAQ$_{+GPTQ}$ | **3.47** | **4.41** |

## D. More Evaluation on Stability of Null Space

**Stability of Null Space under Different Calibration and Inference Distributions.** We sample two batches of input data from two different datasets, WikiText2 and C4, and evaluate the discrepancy between the corresponding null spaces. To quantify this, we examine the principal angles between the null spaces $\mathcal{N}_1$ and $\mathcal{N}_2$ induced by the two input batches across

different layers. The cosine of each principal angle corresponds to a singular value of $\mathcal{N}_1^\top \mathcal{N}_2$; the closer these singular values are to 1, the more aligned the subspaces are. The results are presented in Table 11. It is demonstrated that the null-space property still holds when the calibration and inference distributions differ.

*Table 11.* Evaluation of null space stability via maximum singular value of $\mathcal{N}_1$ and $\mathcal{N}_2$ when calibration and inference distributions differ.

| Layer | Layer 0 | | | Layer 7 | | |
|---|---|---|---|---|---|---|
| | k_proj | q_proj | v_proj | k_proj | q_proj | v_proj |
| Max singular value of $\mathcal{N}_1^\top \mathcal{N}_2$ | 0.967 | 0.977 | 0.973 | 0.970 | 0.969 | 0.968 |

**Stability of Null Space under Different Calibration Set Sizes.** We further evaluate the impact of different calibration set sizes on the null space. Table 12 reports the W3A16 quantization results of GPTQ+OSAQ under varying calibration set sizes. As the calibration set size increases from 128, the perplexity changes only marginally, indicating that the null space remains consistently stable across different calibration set sizes.

*Table 12.* Evaluation of null space stability under different calibration set sizes.

| Model | Calibration size | WikiText2 | C4 |
|---|---|---|---|
| | 64 | 6.84 | 8.81 |
| | 128 | 6.75 | 8.70 |
| LLaMA2-7B | 256 | 6.74 | 8.70 |
| | 512 | 6.72 | 8.69 |
| | 1024 | 6.72 | 8.69 |

# E. Quantization Runtime

Table 13 reports the quantization runtime on an Nvidia A100 GPU. Compared with GPTQ, OSAQ+GPTQ incurs only a marginal overhead (e.g., 24 min vs. 22 min on LLaMA2-7B). This is because the proposed OSAQ obtains the coefficient matrix in closed form, without requiring iterative optimization or additional training.

*Table 13.* Quantization runtime on an Nvidia A100 GPU. As shown, OSAQ incurs only a marginal overhead.

| Method | 2-7B | 2-13B | 2-70B |
|---|---|---|---|
| GPTQ | 22 min | 40 min | 4.0 hr |
| OSAQ$_{+GPTQ}$ | 24 min | 45 min | 4.6 hr |

# F. Assessment of Inference Speedup

Table 14 reports the per-token generation latency of different models during the decoding stage on an Nvidia A100 GPU. As expected, quantization substantially reduces inference time compared to the FP16 baseline, and W4A16 quantization yields speedups of $1.89\times$, $2.41\times$, and $1.96\times$ on LLaMA2-7B, LLaMA2-13B, and LLaMA3-8B, respectively. Importantly, OSAQ introduces no additional inference overhead, ensuring that the acceleration ratios remain consistent with standard baselines.

*Table 14.* Per-token generation latency for the decoding stage on an Nvidia A100 GPU. As shown, OSAQ does not introduce any additional inference overhead, thus maintaining consistent acceleration ratios.

| Method | 2-7B | 2-13B | 3-8B |
|---|---|---|---|
| FP16 latency | 10.8 ms | 19.1 ms | 12.4 ms |
| W4A16 latency | 5.71 ms | 7.90 ms | 6.29 ms |
| W4A16 Speedup | $1.89\times$ | $2.41\times$ | $1.96\times$ |

