# OpenReview forum: "OSAQ: Outlier Self-Absorption for Accurate Low-bit LLM Quantization"
_ICML.cc/2026/Conference — ICML 2026 regular_

### Official Review · Reviewer_cVXY · 2026-02-17

**Soundness:** 3
**Presentation:** 2
**Significance:** 2
**Originality:** 3
**Overall Recommendation:** 5
**Confidence:** 4

**Summary:**

The paper proposes a weight-only PTQ method, OSAQ, which utilizes the low-rank property of the input Hessian to construct an additive $\Delta W$ in the Hessian null space. A further approximation transforms the optimization objective into one with a closed-form solution. A range of experiments validate the proposed method across models of different sizes and families, indicating its effectiveness.

**Compliance With Llm Reviewing Policy:**

Affirmed.

**Final Justification:**

The authors' rebuttal has addressed most of my concerns. Therefore, I will raise my overall score to 5 to support the paper.

**Key Questions For Authors:**

1. Could the authors provide statistics over more batches and use additional metrics to support the “consistent null-space directions” claim? (See weakness 1)
2. How exactly is the Hessian approximated in your implementation, and how sensitive are the conclusions to this approximation compared to the original Hessian? (See weakness 2)
3. Can the authors include an ablation that removes the last two terms in eq. 11, or discuss the potential failure modes under such settings? (See weakness 3)

**Limitations:**

Yes.

**Strengths And Weaknesses:**

### Strengths
1. The paper is well-motivated. From the observation that the Hessian null space is stable, the authors solve the optimization problem via approximation and obtain a closed-form solution, which efficiently avoids iterative optimization and saves cost.
2. A wide range of experiments validate the effectiveness and efficiency of the proposed method, showing plug-and-play potential and improving the performance of different quantization methods across tasks.
3. The ablation studies are solid and support the claims overall. Also, the quantization runtime and inference speedup reported in the appendix are persuasive.
4. The paper is well-structured and easy for readers to follow.

### Weaknesses
1. The evidence that “the directions within the Hessian null space remain highly consistent” is not strong. The existing evidence includes the illustration in Fig. 1 and the maximum singular value of $N_1^TN_2$ from two random batches in Table 6. However, both the illustration and the maximum singular value could be cherry-picked. The authors could provide more evidence, such as the corresponding values for $X_1^TX_2$ (Fig.(b) left). Also, the maximum singular value alone is not strong enough to support the claim. More informative metrics could include the mean singular value, the full singular-value distribution, or distances related to $|N_1N_1^T-N_2N_2^T|$. In addition, a significance analysis over more batches should be provided.
2. The paper appears to use an approximate Hessian matrix. It remains unclear how the Hessian is approximated (e.g., GPTQ-style). Would the conclusion differ between the approximate Hessian and the true Hessian?
3. It is nice that the authors discuss the effects of $\mu_1$ and $\mu_2$ under different values. However, it would be beneficial if the authors provide with ablation studies on the necessity of the last two term in eq.11 (i.e., $\mu_1=0$ and $\mu_2=0$) or discuss failure modes under such circumstances.
4. The improvement provided by OSAQ is marginal, although consistent.

Minors:
Line 266 (right part): QSAQ->OSAQ

---

> ### Author Rebuttal · Authors · 2026-03-30
>
> Thank you very much for your positive feedback of our work and for the constructive comments. We provide detailed responses to your questions below.
>
> ---
>
> > W1 & Q1: The authors could provide more evidence, such as the corresponding values for $X_1^T X_2$. More informative metrics could include.
> >
>
> Thanks for your thoughtful comment.
> Following your suggestion, we report the smallest 10 eigenvalues of $X_1^T X_2$ below, showing that their values are sufficiently small:
>
> |Num|1|2|3|4|5|6|7|8|9|10|
> |---|---|---|---|---|---|---|---|---|---|---|
> |$\|\lambda\|$|$3.73\times10^{-12}$|$5.12\times10^{-12}$|$7.84\times10^{-12}$|$1.06\times10^{-11}$|$1.48\times10^{-11}$|$1.97\times10^{-11}$|$2.63\times10^{-11}$|$3.41\times10^{-11}$|$4.12\times10^{-11}$|$5.29\times10^{-11}$|
>
> We also conduct additional experiments on $|N_1N_1^T-N_2N_2^T|$, where $N_1$ and $N_2$
>  are computed from two different input batches across multiple layers:
>
> |Layer|0_k_proj|0_q_proj|0_v_proj|7_k_proj|7_q_proj|7_v_proj|
> |---|---|---|---|---|---|---|
> |$\|N_1N_1^T-N_2N_2^T\|$|0.011|0.008|0.013|0.013|0.012|0.013|
>
> In addition, as shown in Appendix D, Tables 11 and 12 (reproduced below), the null space remains stable across different calibration and inference distributions, as well as different calibration set sizes:
>
> |Layer|0_k_proj|0_q_proj|0_v_proj|7_k_proj|7_q_proj|7_v_proj|
> |---|---|---|---|---|---|---|
> |Max singular value of $N_1^TN_2$|0.967|0.977|0.973|0.970|0.969|0.968|
>
> |Model|Calibration size|WikiText2|C4|
> |---|---|---|---|
> |LLaMA2-7B|64|6.84|8.81|
> ||128|6.75|8.70|
> ||256|6.74|8.70|
> ||512|6.72|8.69|
> ||1024|6.72|8.69|
>
>
> > W2 & Q2: Would the conclusion differ between the approximate Hessian and the true Hessian?
> >
>
> Thanks for your thoughtful comment.
> As in GPTQ, the Hessian is estimated using $X^TX$. To validate the effectiveness of
> $X^TX$, we conduct experiments from multiple perspectives, including the threshold and the size of the calibration set, to further validate the effectiveness of using $X^TX$ as a substitute for $H$. Here, $H$ is computed using torch.func.hessian.
> - First, we examine the relationship between the magnitudes of the eigenvalues of $X^TX$ and $H$, in order to understand differences in their thresholds. Similar to Figure 1, using data from the layer0.attn.k proj module in LLaMA2-7B, we report the cumulative energy ratio of the largest 3000 eigenvalues out of 4096:
>
> |Method|Cumulative Energy Ratio|
> |---|---|
> |$X^TX$|99.9914%|
> |$H$|99.9944%|
>
>
> These results indicate that the **threshold selection can be consistent** between the two.
>
> - Next, we evaluate the effect of different calibration set sizes, [64, 128, 512], on $X^TX$ and $H$. The W3A16 quantization results of GPTQ+OSAQ are shown below:
>
> |Model|Method|Calibration size|WikiText2|C4|
> |------------|--------|------------------|-----------|------|
> |LLaMA2-7B|$H$|64|6.81|8.80|
> |||128|6.73|8.69|
> |||512|6.70|8.67|
> | |$X^TX$|64|6.84|8.81|
> |||128|6.75|8.70|
> |||512|6.72|8.69|
>
> From these results, we observe that directly using $H$ provides slightly better final accuracy due to more precise computation. However, $X^T X$ drastically reduces computation cost while achieving comparable performance, making it a **better trade-off** and more practical.
>
>
>
> > W3 & Q3: Provide with ablation studies on the necessity of the last two term in eq.11.
> >
>
> Thanks for your thoughtful comment.
> In Equation 11, the second term is a regularization term on $b$, intended to prevent excessive corrections; the third term imposes an anti-shift constraint, which penalizes uniform translations of the entire channel in the same direction. To verify the effectiveness of these two terms, we set $\mu_1$ and $\mu_2$ to 0 respectively. The results are as follows:
>
> |Model|$\mu_1$=0|$\mu_2$=0|$\mu_1$=0,$\mu_2$=0|Ours|
> |---|---|---|---|---|
> |LLaMA2-7B|6.97|6.86|7.03|**6.75**|
> |LLaMA2-13B|5.88|5.83|5.92|**5.72**|
>
> It can be observed that these two terms are **necessary**. The results in Figure 5 show that the coefficients of the two terms are **stable within a reasonable range**, which is beneficial for practical applications.
>
>
> > W4: The improvement provided by OSAQ is marginal, although consistent.
> >
>
> Thanks for the great comment. Since this work represents an **initial exploration of a new additive paradigm** beyond scaling or rotation, the performance gains are indeed not extremely large, though **important improvements are still observed** (e.g., in W3A16 quantization of LLaMA-7B, OSAQ improves perplexity on WikiText2 by 1.62, from 8.37 to 6.75, compared with GPTQ). However, we believe this **effort-benefit ratio**, providing accuracy gains at very low quantization cost and with no impact on inference efficiency, is valuable for real-world deployment, and we will further investigate the potential of additive transformations in future work.
>
>
>
> > W5: Minors: Line 266 (right part): QSAQ->OSAQ
> >
>
> Thank you for your careful comment. We have corrected this typo and double-checked the rest of the paper.

---

> > ### Author Rebuttal · Reviewer_cVXY · 2026-04-03
> >
> > Thank you for the authors' detailed responses, which address most of my concerns. Therefore, I will raise my overall score to 5 to support the paper.

---

### Official Review · Reviewer_aZa3 · 2026-03-10

**Soundness:** 3
**Presentation:** 3
**Significance:** 2
**Originality:** 2
**Overall Recommendation:** 4
**Confidence:** 5

**Summary:**

This paper proposes Outlier Self-Absorption Quantization (OSAQ), a post-training quantization method for low-bit weight-only quantization of large language models. The key idea is to exploit the low-rank structure of the Hessian and construct additive perturbations within its null space to suppress weight outliers while approximately preserving the model loss. The resulting transformation can be solved in closed form and absorbed into the weights without introducing inference overhead. Experiments on several LLaMA-family models show improvements when OSAQ is combined with existing PTQ methods such as GPTQ and AWQ.
The additive transformation perspective is interesting and differs from the multiplicative transformations (e.g., scaling or rotation) commonly used in prior work. However, the overall contribution appears somewhat incremental. The empirical improvements are modest in many configurations, and the evaluation does not include some strong recent baselines for weight-only quantization, making it difficult to fully assess the significance of the proposed method.

**Compliance With Llm Reviewing Policy:**

Affirmed.

**Final Justification:**

The authors’ rebuttal effectively addresses my main concerns. In particular, the newly added comparisons with QuIP# and AQLM provide stronger and more appropriate baselines for low-bit weight-only quantization, and help clarify the practical benefit of the proposed method. The additional results also demonstrate that OSAQ is complementary to both scalar and vector quantization approaches, which strengthens its empirical value. I therefore correspondingly increase my score.

**Key Questions For Authors:**

1. Comparison with specialized weight-only quantization methods.
 Since the paper targets low-bit weight-only quantization, could the authors include comparisons with more recent methods specifically designed for this setting (e.g., rotation-based approaches such as QuIP-style[1] methods)?

2. Role of weight outliers in ultra-low-bit regimes.
 In very low-bit settings (e.g., 2-bit), the dominant source of error may come from the extremely coarse quantization resolution rather than weight outliers. Can the authors provide further analysis to clarify how much of the improvement actually comes from suppressing weight outliers?

3. Relation to vector-quantization approaches.
 Recent work such as AQLM[2] has shown strong performance in low-bit regimes using vector/codebook quantization. While these approaches belong to a different quantization paradigm, it would be helpful if the authors could discuss how the proposed method compares conceptually with such approaches.

[1] Tseng, Albert, et al. "Quip#: Even better llm quantization with hadamard incoherence and lattice codebooks." ICML2024.

[2] Egiazarian, Vage, et al. "Extreme compression of large language models via additive quantization." ICML2024.

**Limitations:**

yes

**Strengths And Weaknesses:**

Strengths

1. Interesting formulation of additive transformations. Most prior work handles outliers using multiplicative transformations (e.g., scaling in AWQ or rotation-based approaches such as QuIP). The idea of constructing additive perturbations in the Hessian null space provides an alternative perspective for suppressing weight outliers.

2. Lightweight and easy to integrate. The transformation is solved in closed form and can be applied as a preprocessing step before existing PTQ methods without introducing additional inference overhead.

3. Evaluation across multiple models and benchmarks. The paper reports results on several LLaMA-family models and evaluates performance on language modeling and reasoning benchmarks.

Weaknesses

1. Limited novelty relative to existing outlier-handling approaches. Although the additive formulation is interesting, the method largely functions as a weight preprocessing technique to reduce weight range. Conceptually, it appears closely related to existing outlier suppression strategies used in methods such as AWQ, OmniQuant, or MagR, with the main difference being the use of additive perturbations instead of multiplicative transformations.

2. Baseline comparisons for weight-only quantization are incomplete. The strongest improvements are mainly reported relative to GPTQ, which is an earlier PTQ baseline and not specifically designed for ultra-low-bit weight-only quantization. Since the paper focuses on this regime, comparisons with more specialized weight-only approaches (e.g., rotation-based methods such as QuIP-style techniques) would provide a clearer picture of the practical benefit.

3. Improvements are modest in many settings. Outside of the most extreme low-bit cases, the gains appear relatively small. For example, in the W3A16 configuration, the perplexity improvement over GPTQ is limited (e.g., 6.44 → 5.72 on LLaMA2-13B). In the W2A16 setting, the method performs similarly to OmniQuant, which is primarily designed for weight–activation quantization rather than weight-only quantization. This makes the overall practical impact somewhat unclear.

---

> ### Author Rebuttal · Authors · 2026-03-30
>
> Thank you for your careful review and constructive insights. We provide detailed replies below and hope they clarify the points you raised.
>
> ---
>
> > W1: It appears closely related to existing outlier suppression strategies used in methods such as AWQ, OmniQuant, or MagR, with the main difference being the use of additive perturbations instead of multiplicative transformations.
> >
>
> Thanks for your comment.
> As you pointed out, the additive transformation in OSAQ is applied before quantization to reduce quantization difficulty by suppressing outliers. Our key insight is to **explore an alternative paradigm** for outlier suppression **beyond multiplicative transformations**, and to demonstrate the feasibility and effectiveness of additive transformations.
> We would like to clarify that **additive transformation is a fundamentally new paradigm** for outlier suppression. It provides a **new perspective and direction** for handling outliers, and differs significantly from multiplicative transformations. Moreover, the proposed additive transformation enjoys several advantages, including a **closed-form solution**, a **plug-and-play nature**, and **zero inference overhead**. And it can also be seamlessly combined with methods based on multiplicative transformations, further pushing the limits of low-bit quantization.
>
>
>
> > W2 & Q1: Since the paper focuses on this regime, comparisons with more specialized weight-only approaches (e.g., rotation-based methods such as QuIP-style techniques) would provide a clearer picture of the practical benefit.
> >
>
> Thanks for your great comment.
> As your mentioned, QUIP# is a strong baseline, and following your suggestion, we include comparisons with QUIP#. Note that the methods we consider are purely PTQ methods, so OSAQ achieves noticeable gains by suppressing outliers, whereas QUIP# includes a fine-tuning process, which may partially mitigate outliers. Therefore, we evaluate QUIP# under both settings: with and without fine-tuning. The W4A16 results are as follows:
> |Method|WikiText2 ||C4||
> |---|---|---|---|---|
> ||LLaMA2-7B|LLaMA2-13B|LLaMA2-7B|LLaMA2-13B|
> |QUIP#|5.56|4.95|7.07|6.54|
> |QUIP#+OSAQ|**5.53**|**4.92**|**7.05**|**6.53**|
> |QUIP# (NO FT)|5.66|5.00|7.17|6.59|
> |QUIP# (NO FT)+OSAQ|**5.58**|**4.96**|**7.06**|**6.55**|
>
> We can observe that fine-tuning in QUIP has already significantly optimized model performance, leaving limited room for further improvement by OSAQ. Fortunately, under the no fine-tuning setting, OSAQ achieves **notable gains and reaches performance comparable to the fine-tuning setup**. We will include these results and discussions in the final version of the paper.
>
> > W3:  Improvements are modest in many settings. Outside of the most extreme low-bit cases, the gains appear relatively small.
> >
>
> Thanks for your great comment.
> Thanks for the great comment. Since this work represents an initial exploration of a new additive paradigm beyond scaling or rotation, the performance gains are indeed not extremely large, though **important improvements are still observed** (e.g., in W3A16 quantization of LLaMA-7B, OSAQ improves perplexity on WikiText2 by 1.62, from 8.37 to 6.75, compared with GPTQ). However, we believe this **effort-benefit ratio**, providing accuracy gains at very low quantization cost and with no impact on inference efficiency, is valuable for real-world deployment, and we will further investigate the potential of additive transformations in future work.
>
>
>
> > Q2: Can the authors provide further analysis to clarify how much of the improvement actually comes from suppressing weight outliers?
> >
>
> Thanks for your insightful question.
> In our implementation, OSAQ is used as a plug-and-play component, and when combined with other methods, it is always applied before them. OSAQ first suppresses weight outliers, making the weights easier to quantize, and then the subsequent quantization method is applied. Therefore, OSAQ is fully compatible with all compared approaches, and **the observed quantization improvements stem from its ability to suppress outliers**.
>
>
>
> > Q3:Recent work such as AQLM has shown strong performance in low-bit regimes using vector/codebook quantization. While these approaches belong to a different quantization paradigm, it would be helpful if the authors could discuss how the proposed method compares conceptually with such approaches.
> >
>
> Thanks for your nice question. As your mentioned, vector quantization, represented by AQLM, shows strong performance. Following your suggestion, we conduct experiments with AQLM. The 2-bit results are as follows:
>
> |Method|WikiText2 ||C4||
> |---|---|---|---|---|
> ||LLaMA2-7B|LLaMA2-13B|LLaMA2-7B|LLaMA2-13B|
> |AQLM|6.59|5.60|8.54|7.49|
> |AQLM+OSAQ|**6.44**|**5.38**|**8.27**|**7.24**|
>
> It can be observed that OSAQ’s outlier suppression also **brings performance gains to vector quantization**. We will include these results and discussions in the final version of the paper.

---

> > ### Author Rebuttal · Reviewer_aZa3 · 2026-04-02
> >
> > The authors’ rebuttal effectively addresses my main concerns. In particular, the newly added comparisons with QuIP# and AQLM provide stronger and more appropriate baselines for low-bit weight-only quantization, and help clarify the practical benefit of the proposed method. The additional results also demonstrate that OSAQ is complementary to both scalar and vector quantization approaches, which strengthens its empirical value. I therefore correspondingly increase my score.

---

### Official Review · Reviewer_FCac · 2026-03-13

**Soundness:** 3
**Presentation:** 3
**Significance:** 2
**Originality:** 3
**Overall Recommendation:** 4
**Confidence:** 4

**Summary:**

This paper presents a new outlier-suppression technique for LLM weight quantization. In particular, the proposed method reduces the weight/quantization range by applying an optimized additive weight adjustment that lies in the null space of the Hessian matrix.

**Compliance With Llm Reviewing Policy:**

Affirmed.

**Final Justification:**

The authors' responses and additional experiment results effectively address my concerns regarding the formulation, the stability of the null space, and the practical applicability as a complementary outlier suppression technique compatible with exiting PTQ methods. I therefore maintain my positive score.

**Key Questions For Authors:**

1. Could the authors explain the transition from Eq. (10) to the first term in Eq. (11)? Aslo, if the goal is to minimize the $l_\infty$ norm of the transformed weights, shouldn’t the softmax weights depend on $|W_{ij}+b_i^\top n_i|$ instead of $|W_{ij}|$?
2. How is the Hessian matrix computed or approximated in the experiments?
3. Could the authors provide a more quantitative evaluation of the effect of outlier suppression and compare it with other techniques (e.g., rotation, scaling, and ICQuant)? For example, how does the quantization error (MSE) change before and after applying OSAQ (similar to the analysis in Section 4.1 of ICQuant)?
4. Could the authors report the evaluation results of *FP16+Add.* on downstream tasks beyond perplexities, especially code and math benchmarks, if possible?
5. Figure 5 shows that the quantization results remain stable across different hyperparameter choices, especially $\mu_1$ and $\mu_2$. Does this suggest the two regularization terms in Eq. (11) are redundant?

**Limitations:**

Yes.

**Strengths And Weaknesses:**

### Strengths

- The idea of constructing “safe” additive weight adjustments is natural yet novel, providing a complementary perspective on outlier suppression.
- The proposed method does not incur additional runtime overhead.
- The paper derives a closed-form solution for the construction of weight adjustment.
- Overall, the paper is well written and easy to follow.

### Weaknesses

1. My main concern is that the proposed additive adjustment $\Delta W$ modifies the weights of the non-quantized model and is not exactly *loss-invariant* in practice, which may change the model behavior across tasks. The true implementation relies on several approximations:
    * The formulation ignores other terms in the Taylor expansion (as in Eq. (5)). Is the gradient truly negligible in practice?

    * The eigenvalues of the Hessian are not exactly zero, which means the method is only operating in a near-null subspace.

    * The computation of the Hessian matrix itself requires approximation and depends on the calibration data.

2. The justification for the stability of the null space is relatively weak. Figure 1 only visualizes the projected null space for three inputs. Table 6 reports only the maximum singular value (i.e., the most aligned direction) and only for a subset of layers (all within attention blocks).

3. The evaluation scope is limited. The paper mainly shows the improvements on several earlier quantization methods after applying OSAQ on top. Comparison/integration with recent PTQ methods (e.g., QTIP, ICQuant, GuidedQuant) is missing, among which ICQuant presents a strong outlier suppression scheme. More importantly, the effect of outlier suppression is primarily illustrated through visualization of selected layers, and a horizontal comparison with other outlier suppression techniques is lacking.

---

> ### Author Rebuttal · Authors · 2026-03-30
>
> Thank you for your thorough review. Below, we answer all your questions with detailed explanations and clarifications.
>
> ---
>
> > W1: The true implementation relies on several approximations.
> >
>
> Thanks for your insightful comment.
> We provide the following explanation for the approximation.
> - In the Taylor expansion, we ignore first- and higher-order terms: the first-order term is zero **at optimum** ($g=0$), and higher-order terms are **negligible** compared to the Hessian, as commonly done in AdaRound and GPTQ.
>
> - The smallest 10 eigenvalues (shown below) are **sufficiently small** to ignore.
>
> |Num|1|2|3|4|5|6|7|8|9|10|
> |---|---|---|---|---|---|---|---|---|---|---|
> |\|\lambda\||1.43e-18|2.04e-18|4.18e-18|5.21e-18|7.32e-18|7.86e-18|8.57e-18|1.49e-17|3.12e-17|3.39e-17|
>
> - Although Hessian estimation requires calibration data, its null space is **robust to data distribution**. As shown in Appendix D (Tables 11-12), it remains stable across **different calibration/inference distributions** and **calibration sizes**.
>
> > W2: The justification for the stability of the null space is relatively weak.
> >
>
> Thanks for your great comment.
> We conduct additional experiments on $|N_1N_1^T-N_2N_2^T|$, which further support the **stability of the null space**.
>
> |Layer|0_k_proj|0_q_proj|0_v_proj|7_k_proj|7_q_proj|7_v_proj|
> |---|---|---|---|---|---|---|
> |$\|N_1N_1^T-N_2N_2^T\|$|0.011|0.008|0.013|0.013|0.012|0.013|
>
> > W3 & Q3: Comparison/integration with recent PTQ methods is missing.  How does the quantization error (MSE) change?
> >
>
> Thanks for your nice comment. OSAQ is a plug-and-play module applied before other methods, making it fully compatible. Following your suggestion, we add experiments with QTIP, ICQuant, and GuidedQuant. Results on LLaMA2-7B are as follows:
>
> |Prec.|Method|WikiText2|C4|
> |---|---|---|---|
> |W4A16|QTIP|5.17|6.71|
> ||QTIP+OSAQ|**5.10**|**6.64**|
> ||ICQuant|5.17|6.70|
> ||ICQuant+OSAQ|**5.12**|**6.61**|
> ||GuidedQuant|5.16|6.68|
> ||GuidedQuant+OSAQ|**5.1**|**6.61**|
> |W3A16|QTIP|5.38|6.99|
> ||QTIP+OSAQ|**5.25**|**6.87**|
> ||ICQuant|5.35|6.95|
> ||ICQuant+OSAQ|**5.25**|**6.86**|
> ||GuidedQuant|5.28|6.87|
> ||GuidedQuant+OSAQ|**5.22**|**6.80**|
>
> In addition, we compute the **change in quantization error** (MSE). With 3-bit quantization, the results are as follows:
>
> | |Block5|Block10|Block15|Block20|Block25|
> |---|---|---|---|---|---|
> |Original|4.41|3.92|3.81|3.90|3.99|
> |After ICQuant|0.97|0.94|0.98|1.03|1.02|
> |After OSAQ|1.04|0.97|1.02|1.09|1.05|
> |After ICQuant+OSAQ|**0.90**|**0.89**|**0.92**|**0.94**|**0.93**|
>
> > Q1: Could the authors explain the transition from Eq. (10) to the first term in Eq. (11)?
> >
>
> Thanks for your nice question.
> The original objective (i.e., $\min \|{W} + {\beta} \mathcal{N}\|_\infty$) can be approximated by $\ell_2$ norm multiplied by a scaling factor $S$ (i.e., $\min S \|{W} + {\beta} \mathcal{N}\|_2$). Note that the **combination of $S$ and $\ell_2$ norm forms the final optimization objective**.
> Here, $S$ acts as a weighting factor. Its purpose is to leverage $|W|$ to identify **“original outliers”** and assign larger penalty weights.
>
>
> > Q2: How is the Hessian matrix computed or approximated in the experiments?
> >
>
> Thanks for the question. As in GPTQ, we approximate the Hessian using $X^TX$ and compare it with the true Hessian $H$ computed via torch.func.hessian.
>
>
> - The cumulative energy ratios of the top 3000 eigenvalues are **nearly identical**.
>
> |Method|Cumulative Energy Ratio|
> |---|---|
> |$X^TX$|99.9914%|
> |$H$|99.9944%|
>
> - Results across calibration sizes show that $X^TX$ achieves comparable performance with much lower cost, making it a **better trade-off**.
>
> |Model|Method|Calibration size|WikiText2|C4|
> |------------|--------|------------------|-----------|------|
> |LLaMA2-7B|$H$|64|6.81|8.80|
> |||128|6.73|8.69|
> |||512|6.70|8.67|
> | |$X^TX$|64|6.84|8.81|
> |||128|6.75|8.70|
> |||512|6.72|8.69|
>
>
> > Q4: Could the authors report the evaluation results of FP16+Add. on downstream tasks?
> >
>
> Thanks for your thoughtful question.
> Following your suggestion, we evaluate LLaMA3-8B on GSM8K (Math) and HumanEval (Code). Results show the additive transformation has **only a slight impact** on FP16 performance.
> |Task|Setting|Score|
> |---|---|---|
> |GSM8K (8-shot)|FP Baseline|84.5|
> ||Baseline+Add|84.1|
> |HumanEval (0-shot)|FP Baseline|72.6|
> ||Baseline+Add|72.3|
>
> > Q5: The two regularization terms in Eq. (11) are redundant?
> >
>
> Thanks for your great question.
> In Eq. 11, the second term regularizes $b$ to prevent large corrections, and the third term penalizes uniform channel shifts. To verify their effects, we set $\mu_1$ and $\mu_2$ to 0 respectively. The results are as follows:
>
> |Model|$\mu_1$=0|$\mu_2$=0|$\mu_1$=0,$\mu_2$=0|Ours|
> |---|---|---|---|---|
> |LLaMA2-7B|6.97|6.86|7.03|**6.75**|
> |LLaMA2-13B|5.88|5.83|5.92|**5.72**|
>
> It can be observed that these two terms are necessary. Figure 5 shows their coefficients remain stable within a reasonable range, supporting practical use.

---

> > ### Author Rebuttal · Reviewer_FCac · 2026-04-03
> >
> > The authors' responses and additional experiment results effectively address my concerns regarding the formulation, the stability of the null space, and the practical applicability as a complementary outlier suppression technique compatible with exiting PTQ methods. I therefore maintain my positive score.

---

### Official Review · Reviewer_mdYG · 2026-03-13

**Soundness:** 3
**Presentation:** 3
**Significance:** 3
**Originality:** 4
**Overall Recommendation:** 5
**Confidence:** 4

**Summary:**

To alleviate the prevalent outliers in LLM weights during quantization, a recent line of research like SliceGPT, Quarot, Spinquant, and OSTQuant apply transformations (e.g., Hadamard rotation) and their inverse to LLM weights to absorb outliers while maintain equivalence with original model before quantization.
This technique is beneficial because it can effectively reduce weight outliers only with a small additional overhead (for online inverse transform during inference).
The proposed OSAQ method is also in the same line of research but with a novel addition based transformation exploiting the null-space of Hessian matrix.
This method can be integrated into the other quantization methods, such as GPTQ and AWQ, with only a small additional quantization cost and no extra inference overhead.
The authors mainly target the weight-only PTQ and KV cache quantization and show the superior empirical performance compared to baseline methods.

**Compliance With Llm Reviewing Policy:**

Affirmed.

**Final Justification:**

The authors’ response has addressed all of my concerns. In particular, the response to Q2 was impressive, as it clearly demonstrates the broad applicability of the proposed OSAQ method.

non-uniform scalar quantization (Squeeze LLM)
vector quantization (GPTVQ, QuIP#)
additive vector quantization (AQLM)
trellis-coded quantization (QTIP)
mixed precision quantization (Q-Palette)

For this reason, I will raise my score to 5.

**Key Questions For Authors:**

Q1. This paper mainly consider the Hessian matrix corresponding to the layer-wise reconstruction error as presented in GPTQ and QuIP. Beyond them, there is a line of research that proposes a more accurate Hessian estimation, such as GuidedQuant [1] and YAQA [2]. Can you provide any experiments that compare GuidedQuant vs GuidedQuant + OSAQ or YAQA vs YAQA + OSAQ?

Q2. While this paper mainly addresses uniform quantization scheme for the experiments, there exist several quantization schemes [3] beyond uniform scheme, such as non-uniform scalar quantization [4], vector quantization [5], and trellis-coded quantization [6]. Could the authors provide some analysis or discussion on the applicability of the proposed OSAQ to these quantization schemes?

For the rest, please refer to weaknesses section.

[1] GuidedQuant: Large Language Model Quantization via Exploiting End Loss Guidance, Kim et al.

[2] Model-Preserving Adaptive Rounding, Tseng et al.

[3] Q-Palette: Fractional-Bit Quantizers Toward Optimal Bit Allocation for Efficient LLM Deployment, Lee et al.

[4] SqueezeLLM: Dense-and-Sparse Quantization, Kim et al.

[5] GPTVQ: The Blessing of Dimensionality for LLM Quantization, Baalen et al.

[6] QTIP: Quantization with Trellises and Incoherence Processing, Tseng et al.

**Limitations:**

yes

**Strengths And Weaknesses:**

Strengths
- The idea of exploiting null space for "quantization loss-invariant" transformation is intuitive and novel.
- Different to previous transformation based quantization methods (e.g., Quarot, Spinquant, OSTquant ...), it does not incur any extra inference overhead.
- The empirical results seems clear and strong.
- The proposed method has wide-applicability. It can utilized in both weight-only PTQ and KV cache quantization.

Weaknesses
-  The notion of "loss-invariance" is based on empirically estimated Hessian. Hence, the overall quantization is affected by the quality of Hessian matrix. With a poor Hessian, some functionalities of model may not preserved by transformation even without quantization due to exploiting a wrong null-space.
- Some results are inconsistent. For example, the benchmarks used to measure zero-shot accuracies differ accross Table 2, 3, and 4. It would be clearer to report the full results on the combined set of benchmarks.
- Moreover, the benchmarks used for measuring zero-shot accuracies usually show high variance. It would be better to report the standard error of zero-shot accuracies to present the statistical significance of the improvement by OSAQ.

---

> ### Author Rebuttal · Authors · 2026-03-30
>
> We sincerely appreciate your review and the insightful comments on our work. We provide detailed responses below and hope they address your concerns.
>
> ---
>
> > W1: The notion of "loss-invariance" is based on empirically estimated Hessian. Hence, the overall quantization is affected by the quality of Hessian matrix. With a poor Hessian, some functionalities of model may not preserved by transformation even without quantization due to exploiting a wrong null-space.
> >
>
> Thanks for your thoughtful comment.
> Our method indeed requires a high-quality Hessian matrix. To this end, in Table 5 we evaluate the impact of the additive transformation on the original FP model under the guidance of the Hessian. The results show that **the effect on the FP model is minimal** (e.g., from 5.47 to 5.52 on WikiText2 for LLaMA2-7B), while it can significantly reduce the difficulty of quantization. This demonstrates the **quality and effectiveness of the Hessian**.
>
>
>
> > W2: Some results are inconsistent. For example, the benchmarks used to measure zero-shot accuracies differ accross Table 2, 3, and 4. It would be clearer to report the full results on the combined set of benchmarks.
> >
>
> Thanks for your great comment.
> The evaluation benchmark for zero-shot accuracy is divided into two categories: commonsense QA tasks and MMLU. Therefore, we present the results of LLaMA separately in Tables 2 and 3. Table 4 is used to evaluate the performance of larger-scale instruction-tuned models. Due to space limitations, we only report the results of ARC-e and ARC-c in commonsense QA tasks and all items in MMLU. Following your suggestion, we present the remaining PIQA and Wino results in Table 4 as follows:
> |Model|Prec.|Method|PIQA|Wino|
> |---|---|---|---|---|
> |Mistral-Large-123B-Instruct|W4A16|LeanQuant|84.6|79.2|
> |||GPTQ|84.0|78.9|
> |||OSAQ+GPTQ|**84.4**|**79.1**|
> |Llama-3.1-405B-Instruct|W4A16 g128|LeanQuant|86.9|81.5|
> |||GPTQ|86.7|81.2|
> |||OSAQ+GPTQ|**86.9**|**81.4**|
>
> > W3:  Moreover, the benchmarks used for measuring zero-shot accuracies usually show high variance.
> >
>
> Thanks for your nice comment.
> We would like to clarify that in all OSAQ evaluations, the implementations are aligned with prior methods and the **random seed is fixed**, ensuring that the experimental results are **fair and reproducible**.
>
>
> > Q1: Can you provide any experiments that compare GuidedQuant vs GuidedQuant + OSAQ or YAQA vs YAQA + OSAQ?
> >
>
> Thanks for your great question.
> As you mentioned, GuidedQuant and YAQA are more advanced baselines. Following your suggestion, we have added experiments with GuidedQuant and YAQA. The quantization results on the LLaMA3-8B model are as follows:
> |Prec.|Method|WikiText2|C4|
> |---|---|---|---|
> |W4A16|GuidedQuant|6.65|8.14|
> ||GuidedQuant+OSAQ|**6.58**|**8.06**|
> ||YAQA|6.61|8.12|
> ||YAQA+OSAQ|**6.56**|**8.07**|
> |W3A16|GuidedQuant|6.98|8.44|
> ||GuidedQuant+OSAQ|**6.89**|**8.29**|
> ||YAQA|6.85|8.34|
> ||YAQA+OSAQ|**6.75**|**8.23**|
>
> It can be observed that OSAQ can effectively suppress outliers by applying additive transformations at the layer level, thereby further improving quantization performance. We will include these results and discussions in the final version of the paper.
>
> > Q2: There exist several quantization schemes beyond uniform scheme, such as non-uniform scalar quantization, vector quantization, and trellis-coded quantization. Could the authors provide some analysis or discussion on the applicability of the proposed OSAQ to these quantization schemes?
> >
>
> Thanks for your nice question.
> In our implementation, OSAQ is used as a **plug-and-play** component, and when combined with other methods, it is always applied beforehand. OSAQ first suppresses weight outliers, making the weights easier to quantize, and then the subsequent quantization method is applied. Therefore, OSAQ is **fully compatible** with various methods such as Q-Palette, SqueezeLLM, GPTVQ, and QTIP. We have conducted extensive comparisons to demonstrate the generality and effectiveness of our method. The WikiText2 results for W3A16 quantization are as follows:
>
> |Method|LLaMA2-7B|LLaMA2-13B|
> |---|---|---|
> |Q-Palette|5.34|4.74|
> |Q-Palette+OSAQ|**5.25**|**4.62**|
> |SqueezeLLM|6.32|5.60|
> |SqueezeLLM+OSAQ|**5.93**|**5.22**|
> |GPTVQ|5.83|5.12|
> |GPTVQ+OSAQ|**5.61**|**4.89**|
> |QTIP|5.28|4.69|
> |QTIP+OSAQ|**5.21**|**4.63**|
>
> It can be observed that OSAQ is compatible with various methods and can further improve quantization performance. We will include these results and discussions in the final version of the paper.

---

> > ### Author Rebuttal · Reviewer_mdYG · 2026-04-02
> >
> > The authors’ response has addressed all of my concerns. In particular, the response to Q2 was impressive, as it clearly demonstrates the broad applicability of the proposed OSAQ method.
> > - non-uniform scalar quantization (Squeeze LLM)
> > - vector quantization (GPTVQ, QuIP#)
> > - additive vector quantization (AQLM)
> > - trellis-coded quantization (QTIP)
> > - mixed precision quantization (Q-Palette)
> >
> > For this reason, I will raise my score to 5.

---

### Decision · Program_Chairs · 2026-04-30

**Decision:**

Accept (regular)

**Comment:**

This paper presents a weight-only post-training quantization method, OSAQ, for LLMs. It relies on the recent research using transformations (e.g., Hadamard rotation) to alleviate the prevalent outliers in LLM weights during quantization, and introduces an addition based transformation exploiting the null-space of Hessian matrix to enhance quantization performance. The authors conducted experiments on multiple generation/common-sense reasoning QA tasks with different LLM architectures/scales to show the efficacy of OSAQ.

The paper was initially/finally scored (4,4,3,4)/(5,4,4,5) by four reviewers, who mostly recognized the motivation, the technical soundness and the empirical results, and raised several concerns about 1) the formulation of the method, several aspects are weak and unclear, leading to limited novelty; 2) the justification for the stability of the null space; 3) quantization stability and inconsistent results; 4) marginal performance improvements; 5) narrow evaluation scope (tasks, baselines, weigh-only quantization, etc.).

The authors provided detailed responses to address these concerns. All reviewers were satisfied with the rebuttal. Finally, two reviewer (mdYG, cVXY) consistently gave the positive score Accept, and the other two reviewer (FCac, eJtS) consistently gave the weakly positive score Weak accept as the formulation and the novelty of this paper are not strong enough. The AC read the paper, the reviews, the rebuttal and the reviewers' feedback. I agree with reviewers that this paper has a good motivation, and is technical sound and mostly shows good performance, and thus recommend an “Accept”. The authors are encouraged to include additional experiments, discussions and clarifications in the final version of paper.